# Mitigating Over-Squashing in Graph Neural Networks by Spectrum-Preserving Sparsification

Langzhang Liang [1]   Fanchen Bu [2]   Zixing Song [3]   Zenglin Xu [4 5]   Shirui Pan [6]   Kijung Shin [1 2]

## Abstract

The message-passing paradigm of Graph Neural Networks often struggles with exchanging information across distant nodes typically due to structural bottlenecks in certain graph regions, a limitation known as *over-squashing*. To reduce such bottlenecks, *graph rewiring*, which modifies graph topology, has been widely used. However, existing graph rewiring techniques often overlook the need to preserve critical properties of the original graph, e.g., *spectral properties*. Moreover, many approaches rely on increasing edge count to improve connectivity, which introduces significant computational overhead and exacerbates the risk of over-smoothing. In this paper, we propose a novel graph rewiring method that leverages *spectrum-preserving* graph *sparsification*, for mitigating over-squashing. Our method generates graphs with enhanced connectivity while maintaining sparsity and largely preserving the original graph spectrum, effectively balancing structural bottleneck reduction and graph property preservation. Experimental results validate the effectiveness of our approach, demonstrating its superiority over strong baseline methods in classification accuracy and retention of the Laplacian spectrum.

## 1. Introduction and Related Works

Graphs are a fundamental data structure for representing complex relational systems, where nodes signify entities and edges represent interactions (Harary, 1969; Diestel, 2012; Shuman et al., 2013). Their capacity to capture both structural and relational information makes them valuable across diverse fields, including social networks, biological systems, and recommendation engines (Battaglia et al., 2018; Sanchez-Gonzalez et al., 2018; Gilmer et al., 2017; van den Berg et al., 2017; Koh et al., 2024). Graph Neural Networks (GNNs) are a specialized class of neural networks developed to process graph-structured data by utilizing both node features and edge-based structural information (Scarselli et al., 2009; Kipf & Welling, 2017; Velickovic et al., 2018). GNNs iteratively update each node's representation by aggregating information from its neighbors (Gilmer et al., 2017).

Recently, the problem of *over-squashing* in GNNs has gained attention. Over-squashing occurs when information from distant nodes is forced through graph structural bottlenecks, causing message distortion during aggregation (Alon & Yahav, 2021). This issue worsens in deeper networks as the receptive field grows, making it difficult to encode large amounts of information into a fixed-size vector. Consequently, over-squashing limits GNNs' ability to capture long-range dependencies, degenerating performance on tasks that need global context.

Many efforts to address over-squashing have focused on *graph rewiring* techniques. These methods modify the edge set of **input graph** (original graph)—either by adding or reconfiguring edges—to create an **output graph** that enhances connectivity and enables more effective message passing in sparse or bottlenecked areas. The rewired output graph is then used for downstream tasks, such as node classification and graph classification.

Despite advancements, existing graph-rewiring methods have two notable limitations. First, they often substantially modify the edge set and fail to preserve graph structural integrity, especially the Laplacian spectrum. For instance, Delaunay graph-based rewiring (Attali et al., 2024) offers benefits such as reduced graph diameter and lower effective resistance. However, it constructs graphs solely based on node features, completely disregarding the original graph topology. Similarly, graph transformers (Ying et al., 2021; Kreuzer et al., 2021) essentially rely on fully connected graphs, while expander graph-based rewiring

[1]Kim Jaechul Graduate School of Artificial Intelligence, KAIST, Seoul, Republic of Korea [2]School of Electrical Engineering, KAIST, Daejeon, Republic of Korea [3]Department of Engineering, University of Cambridge, Cambridge, United Kingdom [4]Artificial Intelligence Innovation and Incubation Institute, Fudan University, Shanghai, China [5]Shanghai Academy of Artificial Intelligence for Science, Shanghai, China [6]School of Information and Communication Technology, Griffith University, Gold Coast, Australia. Correspondence to: Kijung Shin <kijungs@kaist.ac.kr>.

*Proceedings of the 42nd International Conference on Machine Learning*, Vancouver, Canada. PMLR 267, 2025. Copyright 2025 by the author(s).

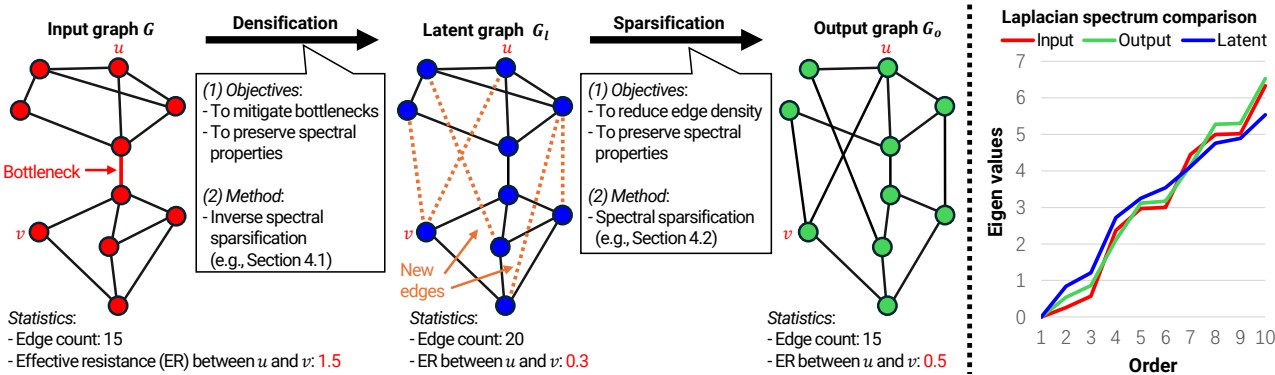

Figure 1: Overview of the proposed densification-sparsification rewiring (DSR) framework. The framework starts by densifying the input graph through an inverse spectral sparsification process, which aims to alleviate structural bottlenecks while preserving spectral properties. Following this, sparsification is applied to the densified graph (referred to as the latent graph) to generate an output graph. This output graph not only exhibits improved connectivity, as measured by effective resistance, but also retains similar spectral properties and maintains the same edge density as the original input graph.

methods (Deac et al., 2022; Christie & He, 2023) may even introduce entirely new node sets, diverging significantly from the structure of the input graph. It is important to preserve graph spectrum during rewiring, since many learning tasks, such as clustering and semi-supervised learning on graphs, rely on spectral properties to ensure accurate results.

Second, these methods often introduce many additional edges to enhance connectivity, which increases the computational cost of learning on the rewired output graph. Moreover, a denser output graph increases the risk of over-smoothing, where information from neighboring nodes becomes excessively blended, undermining the ability to maintain distinct node features (Li et al., 2018; Chen et al., 2020a; Liang et al., 2023b).

To address these challenges, we propose **GOKU**, a graph rewiring method designed to address over-squashing in GNNs. Specifically, given an input graph $G$, the objective is to generate an output graph $G_o$ with improved connectivity by increasing a few of the smallest eigenvalues of its Laplacian, while preserving sparsity and ensuring the remaining eigenvalues stay close to those of $G$. This balance allows $G_o$ to improve message-passing capabilities without deviating significantly from the spectrum of the original graph, which reflects the graph's essential characteristics. This is inspired by spectral sparsification techniques that allow graphs with different edge sets to maintain spectral similarity. GOKU operates in two phases: **(1; densification)** it first densifies the input graph $G$ by solving an inverse graph sparsification problem, creating a densified latent graph with improved connectivity; **(2; sparsification)** it then applies a spectral sparsification algorithm to reduce edge set size, producing a sparse output graph $G_o$ for downstream classification tasks. This densification-sparsification paradigm is illustrated using an example graph in Figure 1.

**Contributions.** Our main contributions are as follows:

- We propose a **novel densification-sparsification rewiring (DSR) paradigm** for mitigating over-squashing (Sec. 3). To our knowledge, DSR is the first rewiring paradigm that simultaneously (1) enhances connectivity for effective message passing, (2) maintains graph sparsity for computational efficiency, and (3) explicitly preserves the graph spectrum for structural fidelity.
- We present **a novel method GOKU, a practical and effective instance of the proposed DSR paradigm** (Sec. 4). With efficient algorithmic designs, GOKU operates in nearly linear time with respect to both nodes and edges.
- We conduct **extensive experiments** (Sec. 5). GOKU achieves better downstream performance than existing methods in both node and graph classification tasks on 10 datasets, while effectively balancing improving connectivity and preserving graph spectrum.

**Related works and their limitations.** Numerous methods have been proposed to mitigate over-squashing in GNNs. Many approaches focus on modifying the graph structure through rewiring. For instance, First-order Spectral Rewiring (FoSR) (Karhadkar et al., 2023) improves the first-order approximation of the spectral gap by adding edges. Curvature-based methods (Topping et al., 2022; Nguyen et al., 2023) optimize connectivity by adding and removing edges based on geometric principles. ProxyGap (Jamadandi et al., 2024) employs both edge deletions and additions to optimize the spectral gap, based on the Braess paradox (Braess, 1968). Expander graph-based rewiring (Deac et al., 2022) maintains a small diameter using Cayley graphs, which may even have a different node set from the original graph. More recently, methods like Probabilistically Rewired Message-Passing Neural Networks (Qian et al., 2024) explore probabilistic approaches to rewiring. Black et al. (2023) propose

minimizing total effective resistance to reduce bottlenecks. Attali et al. (2024) construct new graphs based on node features, completely discarding the original topology. Other methods address over-squashing without altering the original graph topology. Giovanni et al. (2023) investigates the impact of model width, depth, and topology. (Tortorella & Micheli, 2022) and PANDA (Choi et al., 2024) propose alternative message-passing mechanisms, with PANDA focusing on expanded width-aware message passing. Cooperative Graph Neural Networks (Finkelshtein et al., 2024) also introduce learned cooperative mechanisms in message passing that can help alleviate information bottlenecks.

While these approaches improve connectivity, they often substantially change the topology, overlooking the significance of retaining the graph structure. To address this limitation, Locality-aware rewiring (Barbero et al., 2024) has been recently proposed to restrict new connections to $k$-hop neighbors, avoiding arbitrary long-range modifications. However, there still remains a lack of methods that explicitly integrate spectrum preservation—a key aspect that captures essential graph structural properties—into mitigating over-squashing. To bridge this gap, this paper introduces a novel approach leveraging spectral sparsification methods, which produce sparsified graphs that explicitly maintain spectral similarity to the original graph. Our code is available at https://github.com/Jinx-byebye/GOKU.

## 2. Preliminaries and Background

### 2.1. Preliminaries on Graphs

**Graph and notation.** We define basic notations used in this paper. Let $G = (V, E, w)$ be an undirected weighted graph, where $V$ is the node set with $n = |V|$, $E$ is the edge set with $m = |E|$, and $w : E \rightarrow \mathbb{R}^+$ assigns positive weights to edges. Note that unweighted graphs are treated as special cases of weighted graphs with all weights being one. Nodes have a feature matrix $X \in \mathbb{R}^{n \times d}$. The weighted adjacency matrix $A \in \mathbb{R}^{n \times n}$ has $A_{ij} = w_{ij}$ if $(v_i, v_j) \in E$, otherwise $A_{ij} = 0$. The degree matrix $D$ is diagonal with $D_{ii} = \sum_{j=1}^{n} A_{ij}$, and the Laplacian matrix is $L = D - A$. For node classification, the label matrix $Y \in \{0, 1\}^{n \times c}$ encodes node label assignments across $c$ classes. For graph classification, the dataset comprises $\{(G_i, y_i)\}_{i=1}^{N}$, where each graph $G_i = (V_i, E_i)$ is labeled $y_i$.

**Graph spectrum.** The graph spectrum of a graph is the set of eigenvalues $\lambda_1 = 0 \leq \lambda_2 \leq \cdots \leq \lambda_n$ of its Laplacian $L$. It is a "fingerprint" that encodes topological characteristics, e.g., node centrality, community structure, and clustering coefficient (Chung, 1997). Since spectral properties are invariant to graph isomorphisms, they provide a robust framework for comparing and analyzing graphs. Preserving spectral similarity is essential for tasks relying

| Task | Dataset | Same Class | Different Class |
|---|---|---|---|
| Graph | IMDB | 0.59 | 0.21 |
| | Mutag | 0.58 | 0.23 |
| | Proteins | 0.13 | 0.05 |
| | Enzymes | 0.05 | 0.03 |
| Node | Cora | 0.2817 | 0.0736 |
| | Citeseer | 0.0013 | 0.0001 |
| | Pubmed | 0.3383 | 0.1121 |

Table 1: Average spectral similarity between graphs and nodes within the same class and across different classes. For the graph-level (IMDB, Mutag, Proteins, Enzymes), similarity is computed based on eigenvalue distributions using the fastdtw algorithm (as they have different numbers of nodes). For the node level (Cora, Citeseer, Pubmed), similarity is measured as the average cosine similarity between node feature vectors derived from eigenvectors.

on these intrinsic graph and node characteristics. For graph classification, graphs belonging to the same class tend to exhibit more similar eigenvalue distributions than graphs from different classes. For node classification, nodes from the same class exhibit higher average cosine similarity in their eigenvector-based feature representations compared to nodes from different classes (using components from the first 128 eigenvectors, resulting in a 128-dimensional feature vector per node). The results supporting these observations for both tasks are summarized in Table 1.

**Spectral gap and effective resistance.** The spectral gap (or algebraic connectivity) of a graph (Chung, 1997; Fiedler, 1973) is the second-smallest (i.e., smallest non-zero) eigenvalue $\lambda_2$ of its Laplacian matrix.[1] A **small** spectral gap implies many structural bottlenecks, i.e., **poor** connectivity. The effective resistance (ER) $R_{u,v}$ (Doyle & Snell, 1984) between nodes $u$ and $v$, quantifies the difficulty of flow between the two nodes: $R_{u,v} = (e_u - e_v)^\top L^+ (e_u - e_v)$, where $e_u$ and $e_v$ are indicator vectors for $u$ and $v$, and $L^+$ is the Moore-Penrose pseudoinverse of the Laplacian $L$. A **lower** $R_{u,v}$ indicates more alternative short paths between $u$ and $v$, implying $(u, v)$ is **well-connected**.

**Graph Neural Networks (GNNs).** GNNs update node representations iteratively by aggregating information from neighboring nodes: $h_v^{(l)} = U_l \left( h_v^{(l-1)}, M_l \left( h_v^{(l-1)}, \{ h_u^{(l-1)} \mid u \in \mathcal{N}(v) \} \right) \right)$, where $M_l$ and $U_l$ are the message and update functions at layer $l$, and $h_v^{(l)}$ are the node embedding of $v$ at layer $l$. For graph classification, we use a readout function $\hat{y} = R \left( \{ h_v^{(L)} | v \in V \} \right)$ to aggregate all node embeddings.

---

[1]We define *spectral gap* following, e.g., Karhadkar et al. (2023), while it can be alternatively defined as the difference between the two largest eigenvalues of the adjacency matrix.

## 2.2. Graph Sparsification

Graph Sparsification (Benczúr & Karger, 1996; Spielman & Srivastava, 2008; Spielman & Teng, 2011) is a family of techniques aimed at approximating a graph $G$ with a sparse graph $H$, ensuring that $H$ is an efficient computational proxy for $G$ while minimizing approximation errors.

**Spectral sparsification.** Among these techniques, *spectral sparsification* produces a sparsifier of the original graph whose Laplacian quadratic form closely approximates that of the original graph across all real vector inputs.

**Definition 2.1** (Spectrally Similar Graphs). Let $G = (V, E, w)$ and $H = (V, E_H, w_H)$ be undirected weighted graphs. We say $H$ is $(1 \pm \epsilon)$-**spectrally similar** to $G$ for an *approximation error parameter* $\epsilon \in (0, 1)$, denoted by $H \overset{1\pm\epsilon}{\approx} G$, if the Laplacian matrices $L_H$ and $L$ (of $H$ and $G$ respectively) satisfy the following inequality for all $x \in \mathbb{R}^n$:

$$(1 - \epsilon)x^\top L x \leq x^\top L_H x \leq (1 + \epsilon)x^\top L x. \qquad (1)$$

*Remark* 2.2. The definition of $x^\top L x = \sum_{(u,v)\in E} w_{uv}(x_u - x_v)^2$ intrinsically connects to various topological characteristics of the graph. For instance, by choosing $x$ to be an indicator vector for a set of nodes $S \subseteq V$, i.e., $x_u = 1$ if $u \in S$ and $x_u = 0$ if $u \notin S$, the quadratic form $x^\top L x$ becomes the weight of the graph cut separating $S$ from its complement $\bar{S}$, denoted $\text{cut}(S, \bar{S})$. That is, $x^\top L x = \text{cut}(S, \bar{S})$. Similarly, other choices of $x$ can reveal connections to node degrees or relationships between eigenvalues. This comprehensive measure allows us to assess the similarity of graphs based on their fundamental structural properties.

**Definition 2.3** (Spectral Sparsifier). Let $G = (V, E, w)$ and $H = (V, E_H, w_H)$ be undirected weighted graphs. $H$ is called a **sparsifier** of $G$ if $E_H \subseteq E$. If a sparsifier $H$ is also $(1 \pm \epsilon)$-spectrally similar to $G$, we say $H$ is a $(1 \pm \epsilon)$-**spectral sparsifier** of $G$.

A common approach to sparsification (i.e., constructing a sparsifier) involves repeating the following process for $q$ rounds: (1) sampling an edge $e_i \in E$ with replacement according to a probability distribution $\{p_e\}_{e\in E}$, and (2) incrementing its weight by $\frac{w_{e_i}}{qp_{e_i}}$. This process ensures that the expected weight of each edge $e$ in the sparsified graph $H$ matches its original weight in $G$, since $qp_e \cdot \frac{w_e}{qp_e} = w_e$. Consequently, the total edge weight is also preserved in expectation. Different sparsification methods primarily differ in their choice of $\{p_e\}$. Our rewiring paradigm builds upon this sparsification framework, which we introduce in the following section.

# 3. DSR: Proposed Rewiring Paradigm

## — A General Framework

In this section, we introduce a novel Densification-Sparsification Rewiring (DSR) paradigm to mitigate over-squashing in GNNs, consisting of two sequential modules: *graph densification* and *graph sparsification*. We denote the **input graph** (original graph) as $G$, the intermediate graph after densification (referred to as the **latent graph**) as $G_l$, the final **output graph** after sparsification as $G_o$, i.e., $G \xrightarrow{\text{densification}} G_l \xrightarrow{\text{sparsification}} G_o$. Below, we first explain the motivations behind these two steps, and then discuss their details. This section introduces the general paradigm, while Section 4 presents a concrete instance of DSR.

## 3.1. Motivations

**Densification.** Our core idea is to view the input graph $G = (V, E)$ as a spectral sparsifier of an unknown latent graph $G_l = (V, E_l)$ that has good connectivity, and we aim to reconstruct $G_l$. In this view, the structural bottlenecks observed in $G$ arise due to the omission of certain critical edges during the sparsification process from $G_l$ to $G$. By reconstructing $G_l$, we aim to recover these "missing edges", which can help mitigate over-squashing and improve the overall graph structure. How can we identify those "missing edges" and reconstruct $G_l$ from the "sparsified" graph $G$?

This question defines the **densification** (or **inverse sparsification**) problem: given a graph $G$, we aim to reconstruct the latent graph $G_l$ such that $G_l$ is the most likely (as in Maximum Likelihood Estimation) latent graph from which a spectral sparsification algorithm produces $G$. Since $G$ is a spectral sparsifier of $G_l$, $G_l$ is expected to effectively preserve the spectrum of $G$ while improving connectivity.

To reintroduce missing edges crucial for increasing connectivity, we consider the inverse process of **unimportance-based spectral sparsification (USS)**. In USS, crucial edges tend to be removed, while less crucial edges tend to remain. Therefore, if $G$ is the result of applying USS to $G_l$, the "remaining edges" in $G$ are less crucial, while the "missing edges" that are in $G_l$ but not in $G$ are supposed to be crucial for connectivity, which we aim to recover.

**Sparsification.** A drawback of densification is the increased computational complexity due to added edges in $G_l$. To balance connectivity and efficiency, we apply **importance-based spectral sparsification** (ISS) to $G_l$, pruning edges that are less critical for connectivity. As a result, the output $G_o$ almost retains $G_l$'s connectivity level while preserving the sparsity and spectrum of $G$, making it an ideal proxy for downstream tasks on $G$.

**Combination.** When we combine the two steps, both $G$ and $G_o$ are spectral sparsifiers of $G_l$. Therefore, the

***spectrum is explicitly preserved*** from $G$ to $G_o$:

$$G_o \overset{1\pm\epsilon}{\approx} G_l; \quad G_l \overset{1\pm\epsilon}{\approx} G \implies G_o \overset{(1\pm\epsilon)^2}{\approx} G.$$

At the same time, by (1) recovering edges that are crucial for connectivity in densification and (2) removing edges that are less crucial in sparsification, we ***maintain sparsity*** and ***enhance the connectivity*** from $G$ to $G_o$.

### 3.2. Detailed Descriptions

**Densification.** The graph densification problem is formulated as a Maximum Likelihood Estimation (MLE) problem.

**Definition 3.1** (Graph Densification Problem). Given $G = (V, E, w)$ and a spectral sparsification algorithm $\phi(\cdot)$ that outputs a $(1 \pm \epsilon)$-spectral sparsifier (see Def. 2.1),[2] the densification problem aims to find a latent graph $G_l = (V, E_l, w_l)$ to maximize the likelihood of $G$ being the result of applying $\phi(\cdot)$ to $G_l$. Formally,

$$G_l = \underset{G_d = (V, E \cup E', w_d)}{\arg\max} \mathbb{P}(\phi(G_d) = G) \tag{2}$$

subject to $E \cap E' = \emptyset,\ G_d \overset{1\pm\epsilon}{\approx} G.$

Here, $E$ is the edge set of the observed sparsifier $G$, and $E'$ is the set of "missing edges" removed during sparsification.

**Unimportance-based spectral sparsification (USS).** As mentioned in Section 3.1, in USS, edges that are crucial for connectivity tend to be removed, while less crucial edges tend to remain. Therefore, the distribution $p_e$ for sampling edges (see Section 2.2) in $\phi(\cdot)$ should give higher $p_{e,\phi}$ values to edges that are less crucial for connectivity. A natural approach is to set $p_e$ inversely proportional to the topological significance of $e$: $p_{e,\phi} \propto \frac{1}{\text{Topological significance of } e}$, where "topological significance" can be quantified using edge importance metrics, e.g., effective resistance or edge betweenness. We will provide detailed algorithmic choices in our practical instance of densification in Section 4.1.

**Sparsification.** In graph sparsification, we use another spectral sparsification algorithm $\psi(\cdot)$ that outputs a $(1 \pm \epsilon)$-spectral sparsifier, to selectively retain edges with high topological significance in the latent graph $G_l$. By doing so, we reduce structural complexity, preserve the graph spectrum, and maintain the enhanced connectivity in $G_l$.

**Importance-based spectral sparsification (ISS).** As mentioned in Section 3.1, in ISS (which is the opposite of USS), edges that are crucial for connectivity tend to remain. Therefore, the distribution $p_{e,\psi}$ in $\psi(\cdot)$ should give

---

[2]The existence of such $\phi(\cdot)$ is guaranteed by existing results. For example, Benczúr & Karger (1996) and Spielman & Srivastava (2008) proposed algorithms to construct a $(1 \pm \epsilon)$-spectral sparsifier with $O(n \log n / \epsilon^2)$ edges for any graph.

---

**Algorithm 1** USS Algorithm

---
1: **Input:** Graph $G_d = (V, E_d, w_d)$ and sampling count $q$
2: **Output:** Graph $G_s = (V, E_s, w_s)$
3: Compute the Fielder vector $f$ (the eigenvector corresponding to the second-smallest eigenvalue $\lambda_2$ of Laplacian) and node degrees $\deg(v)$'s
4: Initialize $w_s(e) \leftarrow 0, \forall e \in E_d$
5: **for** $i = 1$ to $q$ **do**
6:     Sample an edge $e_i = (u, v) \in E_d$ with replacement, with probability $p_e \propto \left( \frac{\deg(u) + \deg(v) + 1}{|f_u - f_v|} \right)$
7:     Increment the weight: $w_s(e_i) \leftarrow w_s(e_i) + \frac{w_d(e_i)}{p_e q}$
8: **end for**
9: **Return** $G_s = (V, E_s, w_s)$ by collecting the sampled edges and their weights: $E_s = \{e \in E_d : w_s(e) > 0\}$

---

higher $p_e$ values to edges that are more crucial. We can similarly set $p_e$ proportional to the topological significance of $e$: $p_{e,\psi} \propto$ Topological significance of $e$, and the "topological significance" in USS and ISS can be defined differently. We will provide detailed algorithmic choices in our practical instance of sparsification in Section 4.2.

**Comparison to existing methods.** Our method differs from existing approaches in several ways. **First**, it explicitly preserves spectrum through (inverse) spectral sparsification. **Second**, it integrates both densification and sparsification. While prior works (Topping et al., 2022; Nguyen et al., 2023; Fesser & Weber, 2023; Jamadandi et al., 2024) also modify edges to mitigate over-smoothing or enhance connectivity, our approach guarantees that the final sparsified graph never exceeds the original graph's density. **Third**, rather than deterministically selecting edges based on predefined connectivity metrics, our method employs a probabilistic edge modification strategy. **Finally**, leveraging graph spectral properties, the number of added edges $|E'|$ is derived from a well-established theoretical result (see Theorem 4.1) rather than being a heuristically chosen value.

## 4. GOKU: Proposed Method
### — *A Practical Instance of DSR*

In this section, we present a practical instance of the Densification-Sparsification Rewiring (DSR) paradigm, termed GOKU (**G**raph rewiring to tackle **O**ver-squashing by **K**eeping graph spectra thro**U**gh spectral sparsification). Implementing the DSR paradigm requires designing USS and ISS algorithms and solving the MLE problem in Eq. (2).

### 4.1. Graph Densification: USS and the MLE Problem

For graph densification, we design an algorithm for the USS method, and propose an efficient procedure to construct the latent graph for the MLE problem in Eq. (2).

**USS instance.** The USS method should assign **low prob-**

**abilities** $p_e$ to edges **crucial** for maintaining connectivity. We design Algorithm 1 as our USS method, which preferentially retains edges with a high value of $\frac{\deg(u)+\deg(v)+1}{|f_u-f_v|}$, where $f_u$ and $f_v$ are components of the Fiedler vector (the eigenvector associated with the second-smallest eigenvalue $\lambda_2$ of Laplacian; see Section 2.1).

Why does a lower value of $\frac{\deg(u)+\deg(v)+1}{|f_u-f_v|}$ indicate higher significance for connectivity? The intuition is: a large $|f_u - f_v|$ suggests weak connectivity between $u$ and $v$, and small degrees imply they are weakly connected to the remaining part of the graph. Removing such edges creates bottlenecks, making them critical for preserving graph connectivity.

Algorithm 1 provably produces spectral sparsifiers.

**Theorem 4.1** (USS Gives Spectral Sparsifiers). *Let $G_d = (V, E_d, w_d)$ and $G_s = (V, E_s, w_s)$ be the input and output of Algorithm 1, with Laplacian $L$ and $\tilde{L}$, respectively. For any $x \in \mathbb{R}^n$, define $\kappa = \kappa(x) = \frac{\|Cx\|_\infty^2}{p_{min}\|Cx\|_2^2}$, where $p_{min} = \min_{e \in E_d} p_e$, and $C \in \mathbb{R}^{2|E_d| \times |V|}$ denotes the signed incidence matrix of $G_d$.[3] For any $\epsilon \in (0, 1)$, if $q \geq \frac{\kappa^2}{2\epsilon^2} \log 8$, then with probability at least $3/4$,*

$$(1 - \epsilon)x^T L x \leq x^T \tilde{L} x \leq (1 + \epsilon)x^T L x.$$

*Proof.* Proof is deferred to Appendix D.1. □

**Solve the MLE by inverse sparsification.** We now solve the MLE problem in Eq. (2) using Algorithm 1 as the USS method, i.e., $\phi(\cdot)$ in Section 3.2. We treat the input graph $G$ as the observed sparsifier, representing the output of Algorithm 1, and aim to find the latent graph $G_l$ as the most likely input to Algorithm 1 that would produce $G$.

However, solving this MLE problem presents several challenges. First, the search space of possible $G_d$'s is intractable, as it includes all supergraphs of $G$ and all possible edge weight assignments. Second, even for a given $G_d$, computing the exact probability $\mathbb{P}(\phi(G_d) = G)$ is difficult.

To address these challenges, we introduce a practical procedure with simplifications and approximations to efficiently obtain the latent graph $G_l = (V, E_l, w_l)$. For the first challenge, to reduce search space, we assume uniform edge weights in $G_l$, i.e., $w_l(e_1) = w_l(e_2)$ for any $e_1, e_2 \in E_l$, focusing on optimizing the edge set. For the second challenge,

---

[3]Each edge $(u, v) \in E_d$ is represented by two rows: one with $C_{k_1 u} = 1$, $C_{k_1 v} = -1$, and zeros elsewhere; and another with $C_{k_2 u} = -1$, $C_{k_2 v} = 1$, and zeros elsewhere.

to simplify calculation, we approximate $\mathbb{P}(\phi(G_d) = G)$ as:

$$
\begin{aligned}
&\mathbb{P}(\phi(G_d) = G) \\
&\approx \prod_{e \in E} (\mathbb{P}(e \in \phi(G_d)))^{w_e} \prod_{e' \in E'} (\mathbb{P}(e' \notin \phi(G_d)))^{\bar{w}} \\
&= \prod_{e \in E} (1 - (1 - p_e)^q)^{w_e} \prod_{e' \in E'} ((1 - p_{e'})^q)^{\bar{w}}, \quad (3)
\end{aligned}
$$

where $\bar{w} = \sum_{e \in E} w_e/|E|$ is the average edge weight, and recall $E_d = E \cup E'$. In Eq. (3), we simplify the formula by (1) assuming edge independence, and (2) using edge weights directly as power exponents (or equivalently, linear weights in the log probability). Since we assume uniform edge weights in $G_l$, we omit $w_l$ in the formula.

With the above simplifications and approximations, now, optimization over $G_d$ is equivalent to optimization over the "missing edges" $E'$, i.e., we aim to find $E' = \text{argmax}_{E^*} \prod_{e \in E} (1 - (1 - p_e)^q)^{w_e} \prod_{e' \in E^*} ((1 - p_{e'})^q)^{\bar{w}}$. Below, we discuss the detailed procedure to find such $E'$.

First, we select an approximation error parameter $\epsilon \in (0, 1)$, set $x$ as the leading eigenvector (the eigenvector associated with the largest eigenvalue $\lambda_n$) of the Laplacian of $G$, and compute the corresponding condition number $\kappa = \kappa(x)$ (see Theorem 4.1). We use the leading eigenvector since we aim to improve the small eigenvalues (e.g., the spectral gap $\lambda_2$; see Section 2.1) which are closely related to structural bottlenecks, while preserving the large eigenvalues. The choice of $\epsilon$ ensures that a sufficient number of new edges can be added to $E'$, specifically requiring $|E'| = |E_l| - |E|$ to exceed a predefined threshold $\alpha$. Next, we set $q = \frac{\kappa^2}{2\epsilon^2} \log 8$, and determine $|E_l|$ such that when sampling $q$ random edges with replacement from $E_l$, the subgraph contains exactly $|E|$ distinct edges in expectation. Further details on this process are provided in Appendix E.

Second, after determining $|E_l|$ (and thus $|E'|$), the task reduces to maximizing the objective function in Eq. (3) by choosing $|E'|$ edges from the complement set $E_{\text{comp}} = \{(u, v) : u, v \in V\} \setminus E$, with $\binom{|E_{\text{comp}}|}{|E'|}$ possible combinations, which is still intractable. To improve efficiency, we focus on edges between "promising nodes" through two steps: **(1) Candidate set construction**: Identify the $2j$ nodes with the largest absolute Fiedler values and the $2j$ nodes with the smallest degrees, where $j$ is the smallest integer such that $\binom{j}{2} \geq |E'|$. This gives $\binom{4j}{2} \approx 16|E'|$ edge combinations. Pairs between such nodes are likely to have a small value of $\frac{\deg(u)+\deg(v)+1}{|f_u-f_v|}$, and thus are crucial for connectivity. **(2) Random selection**: Assign each edge $e$ in the candidate set a probability $c_e$ proportional to the value of the objective function Eq. (3) assuming $e$ is inserted into $G$. We then sample $|E'|$ edges from the candidate set at once based on the distribution $\{c_e\}$ and construct $G_l$ by including these edges. Finally, we re-scale the edge weights in $G_l$ so that

each edge in $E_l$ has weight $|E|/|E_l|$, ensuring that $G_l$ and $G$ have the same total edge weight, as the total edge weight is typical preserved in sparsification (see Section 2.2). See Appendix B for more details.

### 4.2. Graph Sparsification: ISS

After deriving $G_l$, we sparsify it using importance-based spectral sparsification (ISS) to obtain the output graph $G_o$. We propose to use effective resistance (ER) to measure edge significance in ISS (see Sec. 2.1 for the definition of ER).

**ISS instance.** We adapt a well-known ER-based sparsification (Spielman & Srivastava, 2008), but further incorporate node features. Specifically, each edge $(u, v)$ is sampled with probability $p_e \propto (1 + S_e)R_e$, where $S_e = (1 + \cos(x_u, x_v))/2 \in [0, 1]$ is the normalized cosine similarity between original node features, and $R_e$ is the edge's ER. Heterophily describes a type of graph where connections predominantly occur between dissimilar nodes. This characteristic has been recognized as a significant challenge for effectively training GNNs across various studies(Zhu et al., 2020; Luan et al., 2022; Liang et al., 2023a; 2024). This prioritizes edges with high ER and feature similarity, thus retaining edges crucial to connectivity while filtering noisy edges with dissimilar node features (as in topological denoising; see, e.g., Luo et al. (2021)). We sample edges until $\beta|E|$ distinct edges are selected, where $0.5 < \beta \leq 1$, to ensure sparsity (specifically, $G_o$ is not denser than $G$). Due to the nature of the sparsification process (see Section 2.2), $G_o$ has the same total edge weight as $G_l$ in expectation. Recall in USS, the total edge weight of $G_l$ matches that of $G$. Therefore, $G_o$ also matches the total edge weight of $G$. The ISS algorithm is similar to Alg. 1, except for the probability distribution $\{p_e\}$. For pseudocodes of our ISS instance and the GOKU method, please refer to Appendix B. Similar to USS, our instance of ISS also provably produces spectral sparsifiers.

**Theorem 4.2** (ISS Gives Spectral Sparsifiers)**.** *Let* $G_d = (V, E_d, w_d)$ *and* $G_s = (V, E_s, w_s)$ *be the input and output of Algorithm 2, with Laplacian* $L$ *and* $\tilde{L}$*, respectively. For any* $\epsilon \in (0, 1)$*, there exists* $q = O(n \log n/\epsilon^2)$*, such that*

$$\forall x \in \mathbb{R}^n, \quad (1-\epsilon)x^T L x \leq x^T \tilde{L} x \leq (1+\epsilon)x^T L x,$$

*holds with high probability.*

*Proof.* Proof is deferred to Appendix D.2. □

### 4.3. Complexity Analysis

**Approximating ER.** Computing ER exactly is computationally intensive, but there are efficient approximation methods in sublinear $O(m \operatorname{poly}(\log n/\delta^2))$ time (Koutis et al., 2014; Peng et al., 2021), where $0 < \delta < 1$ is

the approximation error. We use the technique by Koutis et al. (2014). The approximation error affects the bound by a constant factor, altering the bound in Theorem 4.2 to $(1 \pm (1 + \delta)\epsilon)$. We fix $\delta = 0.1$ throughout our experiments.

**Time complexity.** The time complexity of GOKU has two main components: graph densification and graph sparsification. Denote $|E| = m$ and $|V| = n$. In graph densification, generating approximately $16|E'|$ candidate edges and computing their scores requires $O(|E'|m)$. In graph sparsification, approximating the ER for all edges incurs $O\left(\frac{m \cdot \operatorname{poly}(\log n)}{\delta^2}\right)$, and the ISS sampling step adds $O\left(\frac{n \log n}{\epsilon^2}\right)$. Combining these, the overall time complexity is: $O(|E'|m) + O\left(\frac{m \cdot \operatorname{poly}(\log n)}{\delta^2}\right) + O\left(\frac{n \log n}{\epsilon^2}\right) = O(|E'|m) + \tilde{O}\left(\frac{m}{\delta^2}\right)$ where $\tilde{O}$ omits polylogarithmic factors. The Fiedler and leading vectors can be approximated in linear time, dominated by the other terms.

## 5. Experiments

In this section, we evaluate the proposed method GOKU across various tasks, benchmarking its performance against state-of-the-art methods for node and graph classification. Our analysis is guided by the following research questions:

- **RQ1: Performance.** How does GOKU compare to leading methods in node/graph classification tasks?
- **RQ2: Spectrum preservation.** How well does GOKU preserve the graph spectrum?
- **RQ3: Ablation study.** What are the contributions of densification and sparsification to GOKU's effectiveness?
- **RQ4: Structural impact and efficiency of GOKU.** How does GOKU improve the structure (homophily and connectivity) of graphs and how efficient is GOKU?

**Datasets.** We evaluate our method on 10 widely used datasets for node/graph classification. For node classification, we use Cora, Citeseer (Yang et al., 2016), Texas, Cornell, Wisconsin (Pei et al., 2020), and Chameleon (Rozemberczki et al., 2021), including both homophilic and heterophilic datasets. For graph classification, we use Enzymes, Imdb, Mutag, and Proteins from TUDataset (Morris et al., 2019). Dataset statistics are summarized in Appendix A.

**Experimental details.** We compare the proposed method GOKU with no graph rewiring (i.e., NONE) and six state-of-the-art rewiring methods: SDRF (Topping et al., 2022), FoSR (Karhadkar et al., 2023), BORF (Nguyen et al., 2023), GTR (Black et al., 2023), Delaunay Rewiring (DR) (Attali et al., 2024), and LASER (Barbero et al., 2024). Each method preprocesses graphs before evaluation with

| Method | Cora | Citeseer | Texas | Cornell | Wisconsin | Chameleon | Enzymes | Imdb | Mutag | Proteins | AR |
|---|---|---|---|---|---|---|---|---|---|---|---|
| NONE | $86.7_{\pm 0.3}$ | $72.3_{\pm 0.3}$ | $44.2_{\pm 1.5}$ | $41.5_{\pm 1.8}$ | $44.6_{\pm 1.4}$ | $59.2_{\pm 0.6}$ | $25.5_{\pm 1.3}$ | $49.3_{\pm 1.0}$ | $68.8_{\pm 2.1}$ | $70.6_{\pm 1.0}$ | 6.60 |
| SDRF | $86.3_{\pm 0.3}$ | $72.6_{\pm 0.3}$ | $43.9_{\pm 1.6}$ | $42.2_{\pm 1.5}$ | $46.2_{\pm 1.2}$ | $59.4_{\pm 0.5}$ | $26.1_{\pm 1.1}$ | $49.1_{\pm 0.9}$ | $70.5_{\pm 2.1}$ | $71.4_{\pm 0.8}$ | 5.70 |
| FOSR | $85.9_{\pm 0.3}$ | $72.3_{\pm 0.3}$ | $46.0_{\pm 1.6}$ | $40.2_{\pm 1.6}$ | $48.3_{\pm 1.3}$ | $59.3_{\pm 0.6}$ | $27.4_{\pm 1.1}$ | $49.6_{\pm 0.8}$ | $75.6_{\pm 1.7}$ | $72.3_{\pm 0.9}$ | 4.80 |
| BORF | $87.5_{\pm 0.2}$ | $73.8_{\pm 0.2}$ | $49.4_{\pm 1.2}$ | $50.8_{\pm 1.1}$ | $50.3_{\pm 0.9}$ | $61.5_{\pm 0.4}$ | $24.7_{\pm 1.0}$ | $50.1_{\pm 0.9}$ | $75.8_{\pm 1.9}$ | $71.0_{\pm 0.8}$ | 3.40 |
| DR | $78.4_{\pm 1.2}$ | $69.5_{\pm 1.6}$ | $67.8_{\pm 2.5}$ | $57.8_{\pm 1.9}$ | $62.8_{\pm 2.1}$ | $58.6_{\pm 0.8}$ | - | $47.0_{\pm 0.7}$ | $80.1_{\pm 1.8}$ | $72.2_{\pm 0.8}$ | 4.55 |
| GTR | $87.3_{\pm 0.4}$ | $72.4_{\pm 0.3}$ | $45.9_{\pm 1.9}$ | $50.8_{\pm 1.6}$ | $46.7_{\pm 1.5}$ | $57.6_{\pm 0.8}$ | $27.4_{\pm 1.1}$ | $49.5_{\pm 1.0}$ | $78.9_{\pm 1.8}$ | $72.4_{\pm 1.2}$ | 3.90 |
| LASER | $86.9_{\pm 1.1}$ | $72.6_{\pm 0.6}$ | $45.9_{\pm 2.6}$ | $42.7_{\pm 2.6}$ | $46.0_{\pm 2.6}$ | $43.5_{\pm 1.0}$ | $27.6_{\pm 1.3}$ | $50.3_{\pm 1.3}$ | $78.8_{\pm 1.6}$ | $71.8_{\pm 1.6}$ | 4.20 |
| GOKU | $86.8_{\pm 0.3}$ | $73.6_{\pm 0.2}$ | $72.4_{\pm 2.2}$ | $69.4_{\pm 2.1}$ | $68.8_{\pm 1.4}$ | $63.2_{\pm 0.4}$ | $27.6_{\pm 1.2}$ | $49.8_{\pm 0.7}$ | $81.0_{\pm 2.0}$ | $71.9_{\pm 0.8}$ | 1.90 |

Table 2: Performance of different methods for both node and graph classification datasets using **GCN** as the model. The best and runner-up results are highlighted in yellow and green, respectively. The average ranking (AR) reflects the mean position of each method across all datasets. -: DR requires graphs to have at least 3 nodes, but some graphs from Enzymes have only 2 nodes.

| Method | Cora | Citeseer | Texas | Cornell | Wisconsin | Chameleon | Enzymes | Imdb | Mutag | Proteins | AR |
|---|---|---|---|---|---|---|---|---|---|---|---|
| NONE | $77.5_{\pm 1.2}$ | $59.3_{\pm 1.3}$ | $46.4_{\pm 4.5}$ | $40.2_{\pm 4.8}$ | $42.1_{\pm 3.6}$ | $56.6_{\pm 0.8}$ | $33.5_{\pm 1.3}$ | $67.7_{\pm 1.4}$ | $76.1_{\pm 3.1}$ | $69.5_{\pm 1.4}$ | 6.20 |
| SDRF | $77.2_{\pm 0.8}$ | $59.9_{\pm 1.2}$ | $44.5_{\pm 3.9}$ | $38.2_{\pm 4.1}$ | $43.1_{\pm 1.6}$ | $58.1_{\pm 1.4}$ | $32.4_{\pm 1.3}$ | $69.4_{\pm 1.4}$ | $79.5_{\pm 2.6}$ | $71.4_{\pm 0.8}$ | 5.80 |
| FOSR | $78.2_{\pm 0.8}$ | $61.4_{\pm 1.5}$ | $43.7_{\pm 4.1}$ | $39.4_{\pm 3.8}$ | $43.7_{\pm 2.8}$ | $59.3_{\pm 0.6}$ | $28.8_{\pm 1.0}$ | $70.6_{\pm 1.3}$ | $74.8_{\pm 1.5}$ | $73.7_{\pm 0.8}$ | 5.00 |
| BORF | $77.6_{\pm 0.9}$ | $60.8_{\pm 0.2}$ | $49.9_{\pm 3.4}$ | $39.9_{\pm 3.9}$ | $46.7_{\pm 2.2}$ | $59.2_{\pm 0.4}$ | $31.4_{\pm 1.5}$ | $70.5_{\pm 1.3}$ | $78.2_{\pm 1.6}$ | $71.9_{\pm 1.3}$ | 4.40 |
| DR | $64.6_{\pm 1.6}$ | $50.8_{\pm 2.0}$ | $57.3_{\pm 2.4}$ | $50.1_{\pm 2.7}$ | $50.1_{\pm 3.0}$ | $60.2_{\pm 0.8}$ | - | $64.8_{\pm 0.8}$ | $74.5_{\pm 2.4}$ | $74.3_{\pm 0.8}$ | 4.44 |
| GTR | $78.6_{\pm 1.3}$ | $62.6_{\pm 1.9}$ | $49.5_{\pm 2.9}$ | $39.4_{\pm 2.4}$ | $44.2_{\pm 2.4}$ | $57.1_{\pm 1.2}$ | $28.4_{\pm 1.9}$ | $70.1_{\pm 1.2}$ | $78.5_{\pm 3.5}$ | $73.3_{\pm 0.9}$ | 4.40 |
| LASER | $79.1_{\pm 1.0}$ | $66.5_{\pm 1.3}$ | $46.5_{\pm 4.5}$ | $44.5_{\pm 3.8}$ | $46.1_{\pm 4.6}$ | $59.8_{\pm 2.2}$ | $35.3_{\pm 1.3}$ | $68.6_{\pm 1.2}$ | $76.1_{\pm 2.4}$ | $72.1_{\pm 0.7}$ | 3.40 |
| GOKU | $78.4_{\pm 0.5}$ | $63.6_{\pm 1.3}$ | $60.2_{\pm 2.6}$ | $49.5_{\pm 3.5}$ | $57.6_{\pm 3.1}$ | $62.1_{\pm 0.6}$ | $33.8_{\pm 1.2}$ | $71.3_{\pm 0.9}$ | $78.4_{\pm 2.5}$ | $73.9_{\pm 1.0}$ | 1.80 |

Table 3: Performance of different methods for both node and graph classification datasets using **GIN** as the model.

| Method | Mutag | Imdb | Proteins | Chameleon | Cora |
|---|---|---|---|---|---|
| GOKU-D | $78.0_{\pm 1.8}$ | $48.2_{\pm 1.1}$ | $72.0_{\pm 0.9}$ | $62.7_{\pm 0.7}$ | $86.4_{\pm 0.7}$ |
| GOKU-S | $76.8_{\pm 1.9}$ | $48.8_{\pm 1.1}$ | $70.1_{\pm 1.9}$ | $62.9_{\pm 1.5}$ | $86.4_{\pm 0.6}$ |
| GOKU | $81.0_{\pm 2.0}$ | $49.8_{\pm 0.7}$ | $71.9_{\pm 0.8}$ | $63.2_{\pm 0.4}$ | $86.8_{\pm 0.3}$ |

Table 4: Ablation study. GOKU-D and GOKU-S correspond to densification and sparsification only.

a GCN[4] (Kipf & Welling, 2017), GIN[5] (Xu et al., 2019), and GCNII (Chen et al., 2020b) (refer to Appendix C). Following Nguyen et al. (2023), we prioritize fairness and comprehensiveness over maximizing performance on individual datasets, applying fixed GNN hyperparameters (e.g., learning rate $1e-3$, hidden dimension 64, 4 layers) across all methods. Hyperparameters for rewiring methods are tuned individually. The configuration that yields the best validation performance is selected and tested. Results are averaged over 100 random trials with both the mean test accuracy and the 95% confidence interval reported.

**Hyperparameters of GOKU.** We fix the spectrum approximation error $\epsilon = 0.1$, which determines the value of $q$ as $q = \frac{\kappa^2}{2\epsilon^2} \log 8$ (see Theorem 4.1). For the ER approximation algorithm (Koutis et al., 2014), we set $\delta = 0.1$. Two hyperparameters are fine-tuned based on the validation set: **(1; densification)** $\alpha \in \{5, 10, 15, 20, 25, 30\}$: If the number of edges added during densification computed with the initial $\epsilon = 0.1$, is less than $\alpha$, we iteratively increase $\epsilon$ until this threshold is exceeded. **(2; sparsification)** $\beta \in [0.5, 1.0]$: This parameter scales the size of the output graph $G_o$ relative to the input graph $G$, such that $|E_o| = \beta|E|$. During the sparsification phase, we keep sampling until $\beta|E|$ distinct

---

[4] $h_v^{(l+1)} = \sigma\left(\sum_{u \in N(v)} \frac{w_{uv}}{\sqrt{d_u \cdot d_v}} h_u^{(l)} W^{(l)}\right)$

[5] $h_v^{(l+1)} = \text{MLP}^{(l)}\left((1 + \epsilon) \cdot h_v^{(l)} + \sum_{u \in N(v)} w_{uv} \cdot h_u^{(l)}\right)$

edges are sampled.

**Performance results.** Table 2 and Table 3 summarize the results for GOKU and baselines using GCN and GIN as backbone models, respectively (see Appendix C for the results using GCNII). GOKU achieves superior performance across both node and graph classification tasks when using GCN as the backbone model, especially in the node classification task for heterophilic graphs (such as Texas, Cornell, and Wisconsin). Similarly, with GIN as the backbone, GOKU maintains its strong performance, outperforming baselines in most datasets. Overall, GOKU consistently outperforms other methods, demonstrating its effectiveness across diverse tasks and graph types.

**Visualization of graph spectra.** We visualize the spectra of randomly selected graphs from the Mutag and Proteins datasets before and after rewiring using GTR, DR, and GOKU. For a fair comparison, 10 edges are added to Mutag and 50 to the other datasets ($\alpha = 10$ or $\alpha = 50$ for GOKU). For LASER, we set $p = 0.15$ and maximum hop $k = 3$ to add a similar number of edges. As shown in Figure 2, GTR and DR cause significant eigenvalue deviations, while GOKU best preserves the spectra of the original graphs, even outperforming LASER. This highlights GOKU's ability to maintain spectral fidelity.

**Ablation study.** We provide ablation study in Table 4. The results demonstrate that both densification and sparsification contribute to the effectiveness of GOKU. While GOKU-D (densification only) enhances connectivity more significantly than GOKU, this does not always translate to better performance for downstream tasks. Excessive connectivity can lead to issues such as over-smoothing or disrupting the underlying community structures. On the other hand,

| Dataset | # Edges | Homo. (bef.) | Homo. (aft.) | Time (sec) |
|---------|---------|--------------|--------------|------------|
| S-High  | 20      | 0.8333       | 0.9167       | 0.086      |
| S-Low   | 20      | 0.4000       | 0.4123       | 0.072      |
| M-High  | 245     | 0.7459       | 0.7764       | 0.42       |
| M-Low   | 245     | 0.3452       | 0.3822       | 0.36       |
| L-High  | 80352   | 0.7558       | 0.7822       | 65.26      |
| L-Low   | 80352   | 0.3854       | 0.4325       | 72.24      |

Table 5: (1) Homophily levels before and after applying the GOKU rewiring method, and (2) the average running time (10 runs) for GOKU rewiring across graphs of varying scales.

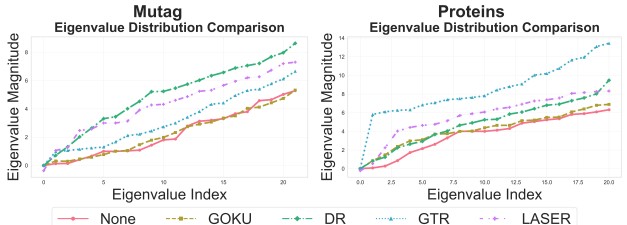

Figure 2: Graph spectra of Mutag and Proteins before and after rewiring. See more examples in Appendix F.2.

GOKU-S (sparsification only) alone fails to achieve significant improvements in connectivity. This underscores the importance of combining both components to strike a balance between connectivity and density.

**Homophily level, effective resistance, and running time.** We generate graphs at varying scales (20, 200, and 2000 nodes) with different homophily levels (high and low) using the Stochastic Block Model (SBM) to investigate the impact of GOKU. Specifically, we compare homophily levels and effective resistance distributions before and after the GOKU rewiring technique is applied to each graph. In addition, we measure the running time of GOKU across different graph sizes to evaluate its computational efficiency as the graph scale increases. For the experiment, we fix $\alpha = 0.1|E|$ and $\beta = 1.0$, meaning that during densification, we add $0.1|E|$ edges, and the density of the rewired graphs remains the same as that of the original graphs.

- **Homophily:** The homophily values of the original and rewired graphs are given in Table 5. Note that GOKU consistently improves homophily levels[6] across different graph scales and homophily levels. This improvement is due to the consideration of node features during sparsification, which helps preserve intra-community edges.

- **Effective resistance:** After applying GOKU, the effective resistances between node pairs significantly decrease, as shown in the distributions in Figure 3. This reduction indicates improvements in both local and global connectivity in the rewired graphs.

- **Running time:** The running time of GOKU scales near-linearly with the graph size, as shown in Table 5,

---

[6]*Homophily level* is defined as $\frac{|\{(u,v)\in E:y_u=y_v\}|}{|E|}$.

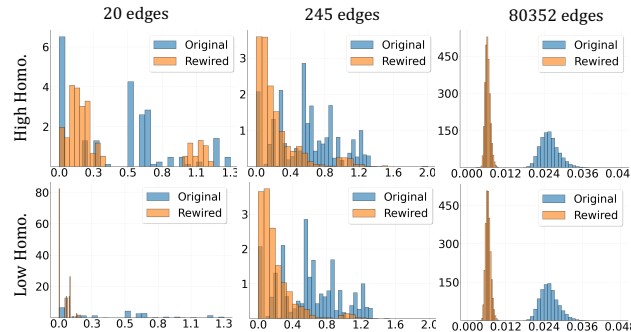

Figure 3: Effective resistance (ER) distribution of all node pairs in graphs with varying scales and homophily levels before and after rewiring. **Smaller** ER values suggest **better** connectivity.

which aligns with the theoretical time complexity in Section 4.3.

## 6. Conclusions and Limitations

In this paper, we propose GOKU, a novel graph rewiring method for addressing over-squashing in GNNs through the densification-sparsification paradigm. Building on spectral sparsification, we formulate and solve an inverse sparsification problem to enhance graph connectivity while preserving spectral properties, followed by a sparsification step for maintaining sparsity. Extensive experiments show that GOKU outperforms existing methods in node and graph classification tasks, while effectively balancing connectivity improvement and spectral retention. Our code is available at https://github.com/Jinx-byebye/GOKU.

**Limitations.** Spectrum preservation for unweighted graphs with different edge sizes is inherently challenging, so spectral sparsification typically relies on weighted graphs, where edge weights may cause additional complexities for GNNs. This is not a major concern in practice for most GNN models, where edge weights can be naturally incorporated.

## Impact Statement

This paper presents work whose goal is to advance the field of Machine Learning. There are many potential societal consequences of our work, none of which we feel must be specifically highlighted here.

## Acknowledgements

We thank the anonymous reviewers for their constructive feedback. This work was partly supported by the National Research Foundation of Korea (NRF) grant funded by the Korea government (MSIT) (No. RS-2024-00406985). This work was partly supported by Institute of Information & Communications Technology Planning & Evaluation (IITP) grant funded by the Korea government (MSIT) (No. 2022-0-00871 / RS-2022-II220871, Development of AI Autonomy and Knowledge Enhancement for AI Agent Collaboration) (No. RS-2024-00438638, EntireDB2AI: Foundations and Software for Comprehensive Deep Representation Learning and Prediction on Entire Relational Databases) (No. RS-2019-II190075, Artificial Intelligence Graduate School Program (KAIST)).

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

---

**Algorithm 2** ISS Algorithm

---

1: **Input:** Graph $G_l = (V, E)$ and sampling count $q$
2: **Output:** Output graph $G_o = (V, E_c)$
3: Approximate ER $R_e$ and compute original node feature cosine similarity $S_e = \frac{1+\cos(x_u, x_v)}{2} \in [0, 1]$ for each edge $e \in E$.
4: **for** $i = 1$ **to** $q$ **do**
5:     Randomly select edge $e$ with probability $p_e \propto (1 + S_e) R_e$ with replacement
6:     Increment the edge weight by $\frac{1}{p_e q}$ for each sampling
7: **end for**
8: Construct the output graph $G_o = (V, E_o)$ by including the sampled edges with their accumulated weights

---

## A. Dataset

A summary of the statistics for all datasets used is presented in Table 6 and Table 7.

|  | **Cornell** | **Texas** | **Wisconsin** | **Cora** | **Citeseer** | **Chameleon** |
|---|---|---|---|---|---|---|
| **#Nodes** | 140 | 135 | 184 | 2485 | 2120 | 832 |
| **#Edges** | 219 | 251 | 362 | 5069 | 3679 | 12355 |
| **#Features** | 1703 | 1703 | 1703 | 1433 | 3703 | 2323 |
| **#Classes** | 5 | 5 | 5 | 7 | 6 | 5 |
| **Directed Graph?** | YES | YES | YES | NO | NO | YES |

Table 6: Summary statistics of node classification datasets.

|  | **ENZYMES** | **IMDB** | **MUTAG** | **PROTEINS** |
|---|---|---|---|---|
| **Basic Info. of Graphs** |  |  |  |  |
| **Nodes (Min–Max)** | 2–126 | 12–136 | 10–28 | 4–620 |
| **Edges (Min–Max)** | 2–298 | 52–2498 | 20–66 | 10–2098 |
| **Avg # of Nodes** | 32.63 | 19.77 | 17.93 | 39.06 |
| **Avg # of Edges** | 124.27 | 193.06 | 39.58 | 145.63 |
| **Classification Info.** |  |  |  |  |
| **# of Graphs** | 600 | 1000 | 188 | 1113 |
| **# of Classes** | 6 | 2 | 2 | 2 |
| **Directed Graphs?** | NO | NO | NO | NO |

Table 7: Summary statistics of graph classification datasets.

## B. The GOKU Algorithm Pseudocode

We provide the pseudocodes for the ISS sparsification and GOKU in Algorithm 2 and Algorithm 3, respectively.

## C. Results with GCNII

We conduct additional experiments using several of the strongest baselines from Table 2 and Table 3, alongside four benchmark datasets: Mutag, Proteins, Cora, and Cornell. In these experiments, we use the powerful GNN model, GCNII (Chen et al., 2020b), as the backbone. The results from these experiments provide valuable insights into the relative strengths of the baseline models and the GCNII backbone. The results in the Table 8 demonstrate the performance of four methods (DR, GTR, LASER, GOKU) on five datasets (Mutag, Proteins, Cora, Cornell, Wisconsin) using GCNII. GOKU consistently achieves the best or near-best performance across most datasets.

---

**Algorithm 3** GOKU, Graph Densification and Sparsification Algorithm

---

1: **Input:** Input graph $G = (V, E)$
2: **Output:** Output graph $G_o = (V, E_o)$
3: {Densification: Add $k$ Edges to $G$}
4: Compute $q = \frac{\kappa^2}{2\epsilon^2} \log 8$ using the leading eigenvector and $\epsilon$; estimate $|E_l|$ as in Appendix E and set $k = |E_l| - |E|$.
5: Identify $2j$ nodes with the largest absolute Fiedler values and $2j$ nodes with the smallest degrees.
6: Form candidate edges between these nodes, size $\binom{4j}{2}$.
7: Initialize $S = \emptyset$ (set of selected edges)
8: Initialize $r = 0$ (sampling counter)
9: **while** size of $S < k$ **do**
10:    Randomly select an edge $e$ from the candidate set based on probability $p_e$ proportional to its contribution to the objective function (3).
11:    **if** $e \notin S$ **then**
12:       Add $e$ to $S$.
13:    **end if**
14:    Increment edge weight of $e$ by $\frac{1}{p_e}$
15:    Increment $r$ by 1.
16: **end while**
17: Construct latent graph $G_l = (V, E_l) = E \cup S$ by including edges in $S$ with their accumulated weights; scale edge weights in $E$ by $|E|/|E_l|$; scale edge weights in $S$ by $1/r$.
18: {Sparsification: Reduce $E_l$ to $E_o$}
19: **for** each edge $e \in E_l$ **do**
20:    Approximate effective resistance $R_e$ with approximation error $\delta = 0.1$ using the technique proposed by Koutis et al. (2014).
21: **end for**
22: **for** $i = 1$ to $q$ **do**
23:    Randomly sample edges with probability $p_e \propto (1 + S_e)R_e$, where $S_e$ and $R_e$ denote feature similarity and effective resistance of edge $e$, respectively.
24:    Increment edge weights by $\frac{1}{p_e q}$ for each sample.
25: **end for**
26: Construct sparsified graph $G_o = (V, E_o)$ by including the sampled edges with their accumulated weights.

---

| Method | Mutag | Proteins | Cora | Cornell | Wisconsin | AR |
|--------|-------|----------|------|---------|-----------|-----|
| DR | $81.6_{\pm 1.7}$ | $72.1_{\pm 1.2}$ | $78.8_{\pm 1.4}$ | $67.9_{\pm 4.7}$ | $70.5_{\pm 3.9}$ | 2.6 |
| GTR | $79.0_{\pm 1.4}$ | $72.5_{\pm 1.1}$ | $86.8_{\pm 1.3}$ | $59.2_{\pm 4.8}$ | $62.7_{\pm 3.7}$ | 3.4 |
| LASER | $80.3_{\pm 1.6}$ | $73.0_{\pm 1.0}$ | $88.1_{\pm 1.3}$ | $60.8_{\pm 3.6}$ | $60.1_{\pm 3.5}$ | 2.6 |
| GOKU | $81.8_{\pm 1.4}$ | $73.9_{\pm 0.9}$ | $87.2_{\pm 1.1}$ | $69.4_{\pm 4.4}$ | $68.8_{\pm 3.6}$ | **1.4** |

Table 8: Results on five datasets with **GCNII**. The best and runner-up results are highlighted in yellow and green, respectively. The average ranking (AR) reflects the mean position of each method across all datasets.

## D. Proof

### D.1. Proof of Theorem 4.1

*Proof.* We aim to show that the Laplacian $\tilde{L}$ of the sparsified graph $G_s$ approximates the Laplacian $L$ of the original graph $G$ in the spectral norm, i.e.,

$$(1 - \epsilon)L \preceq \tilde{L} \preceq (1 + \epsilon)L$$

with high probability. Our approach is based on Chernoff bounds and variance control in the sampling process.

**Setup.** Consider a graph $G = (V, E)$ with $n$ nodes and $m$ edges. For each edge $e = (u, v) \in E$, the weight $w_e = 1$. The Laplacian matrix $L \in \mathbb{R}^{n \times n}$ is defined as

$$L = D - A$$

where $D$ is the degree matrix and $A$ is the adjacency matrix. We sample $q$ edges from $E$ with replacement, where the probability of sampling an edge $e = (u, v)$ is denoted by $p_e$, without specifying a particular distribution for $p_e$ values. For each sampled edge, its contribution to the Laplacian is scaled by the inverse of its sampling probability. Let $\tilde{L}$ be the Laplacian of the sparsified graph. Then,

$$\mathbb{E}[\tilde{L}] = L.$$

We aim to control the spectral deviation between $\tilde{L}$ and $L$ using a concentration bound.

**Quadratic Form Decomposition.** Let $x \in \mathbb{R}^n$ be an arbitrary vector. We analyze the quadratic form $x^T \tilde{L} x$, which is a sum of independent random variables:

$$x^T \tilde{L} x = \sum_{i=1}^{q} X_i,$$

where $X_i$ is the contribution of the $i$-th sampled edge to the quadratic form. More precisely, for each sampled edge $e = (u, v)$, we define:

$$X_i = \frac{1}{qp_e}(x_u - x_v)^2.$$

The random variables $X_i$ are independent and identically distributed, and the expectation of each $X_i$ is:

$$\mathbb{E}[X_i] = \frac{1}{q}x^T L x = \frac{1}{q}\mu,$$

where $\mu = x^T L x$ is the quadratic form of the original Laplacian.

**Bounding the Range of $X_i$.** To apply concentration inequalities, we need to bound the range of the random variables $X_i$. The maximum possible value of $X_i$ occurs when the difference $(x_u - x_v)^2$ is maximal, i.e.,

$$\max_{e=(u,v)\in E}(x_u - x_v)^2 = \|Cx\|_\infty^2,$$

where $C \in \mathbb{R}^{m \times n}$ is the signed incidence matrix of the graph. Therefore, each $X_i$ is bounded as:

$$0 \le X_i \le \frac{1}{qp_{\min}}\|Cx\|_\infty^2,$$

where $p_{\min} = \min_{e \in E} p_e$ represents the minimum sampling probability.

**Variance Bound.** The variance of each $X_i$ is bounded as:

$$\mathrm{Var}(X_i) \le \mathbb{E}[X_i^2] \le \frac{1}{q^2 p_{\min}^2}\|Cx\|_\infty^4.$$

Summing over all sampled edges, the total variance is:

$$\mathrm{Var}\left(\sum_{i=1}^{q} X_i\right) = q \cdot \mathrm{Var}(X_1) \le \frac{q \cdot \|Cx\|_\infty^4}{q^2 p_{\min}^2}.$$

Therefore, the standard deviation is:

$$\sigma \le \frac{\|Cx\|_\infty^2}{q^{1/2}p_{\min}}.$$

**Application of Hoeffding's Inequality.** We now apply Hoeffding's inequality to bound the probability that the sum deviates from its expectation:

$$\mathbb{P}\left(\left|\sum_{i=1}^{q}(X_i - \mathbb{E}[X_i])\right| > t\right) \le 2\exp\left(\frac{-2t^2}{\sum_{i=1}^{q}(b_i - a_i)^2}\right).$$

Substituting the bounds on $X_i$, we get:

$$\mathbb{P}\left(\left|\sum_{i=1}^{q}(X_i - \mathbb{E}[X_i])\right| > t\right) \le 2\exp\left(\frac{-2t^2 q p_{\min}^2}{\|Cx\|_\infty^4}\right).$$

**Conclusion.** Let $t = \epsilon\mu$ and $\kappa = \frac{\|Cx\|_\infty^2}{p_{\min}\|Cx\|_2^2}$. Then the probability becomes:

$$\mathbb{P}\left(|x^T \tilde{L} x - x^T L x| > \epsilon\mu\right) \le 2\exp\left(\frac{-2\epsilon^2 q}{\kappa^2}\right).$$

Thus, by choosing $q \ge \frac{\kappa^2}{2\epsilon^2}\log 8$, we ensure that

$$|x^T \tilde{L} x - x^T L x| \le \epsilon x^T L x$$

with probability at least $3/4$, which completes the proof. $\qquad\square$

### D.2. Proof of Theorem 4.2

*Proof.* This proof builds upon the techniques introduced by Spielman & Srivastava (2008).

Let $C \in \mathbb{R}^{m\times n}$ be a signed incidence matrix defined such that $C_{e,v} = 1$ if $v$ is the head of edge $e$, $C_{e,v} = -1$ if $v$ is the tail of edge $e$, and $C_{e,v} = 0$ otherwise. Consequently, the Laplacian can be expressed as $L = C^T C$. This allows us to denote the Laplacian of the sparsifier as $\tilde{L} = C^T S C$, where $S$ is a diagonal matrix with $S_{e,e} = \frac{s_e}{qp_e}$, and $s_e$ represents the number of times edge $e$ is sampled.

Since $G$ is a connected graph, the multiplicity of the eigenvalue $0$ is $1$, and we express the pseudoinverse of $L$ as

$$L^+ = \sum_{i=2}^{n} \frac{1}{\lambda_i} u_i u_i^T.$$

Using the pseudoinverse, the effective resistance between nodes $u$ and $v$ can be calculated as follows:

$$R_{uv} = (1_u - 1_v)^T L^+ (1_u - 1_v),$$

where the vector $1_u$ is an indicator vector, with the $u$-th element equal to $1$ and all other elements equal to $0$. Based on this definition, we can represent the effective resistance matrix as

$$\Phi = C L^+ C^T,$$

where the diagonal entries are given by $\Phi_{e,e} = R_e$. Next, we introduce the following lemma:

**Lemma D.1.** *Let $p$ be a probability distribution over $\Gamma \subseteq \mathbb{R}^d$ such that for all $y \in \Omega$, $y \le \tau$, and $\|\mathbb{E}_p[yy^T]\| \le 1$. Let $y_1, y_2, \ldots, y_q$ be independent and identically distributed samples drawn from $p$. Then*

$$\mathbb{E}\left\|\frac{1}{q}\sum_{i=1}^{q} y_i y_i^T - \mathbb{E}yy^T\right\| \le \min\left(c\tau\sqrt{\frac{\log q}{q}}, 1\right),$$

*where $c$ is a constant.*

We will apply the above lemma to $\mathbb{E}\|\Phi S\Phi - \Phi\Phi\|$. Note that $\Phi S\Phi$ can be expressed as

$$\Phi S\Phi = \sum_e S_{e,e}\Phi_{:,e}\Phi_{:,e}^T = \frac{1}{q}\sum_e t_e \frac{\Phi_{:,e}}{\sqrt{p_e}}\frac{\Phi_{:,e}^T}{\sqrt{p_e}} = \frac{1}{q}\sum_{i=1}^{q} y_i y_i^T, \tag{4}$$

for $y_i$ drawn independently from the distribution $y = \frac{\Phi_{:,e}}{\sqrt{p_e}}$ with probability $p_e$. We compute the expectation of $yy^T$ as follows:

$$\mathbb{E}yy^T = \sum_e p_e \frac{1}{p_e}\Phi_{:,e}\Phi_{:,e}^T = \Phi^2 = \Phi. \tag{5}$$

The last step follows from the fact that $\Phi$ is a projection matrix, which can be derived as follows:

$$\Phi^2 = CL^+C^TCL^+C^T = CL^+LL^+C^T = CL^+C^T = \Phi. \tag{6}$$

The first equation uses the definition of $L = C^TC$, while the second equation follows from the fact that $L^+L = \sum_{i=2}^n u_i u_i^T$, which indicates that $L^+L$ is an identity on the image of $L^+$, i.e., $L^+LL^+ = L^+$.

Next, to apply Lemma D.1, we calculate $\|\mathbb{E}yy^T\|_2$ and the upper bound of $\|y\|_2$. Since $\Phi$ is a projection matrix, its eigenvalues are either 1 or 0. To determine the multiplicity of 1, consider $\forall y \in \text{im}(C)$, there exists a vector $x \perp \ker(C)$ such that $Cx = y$. We have

$$\Phi y = CL^+C^TCx = CL^+Lx = Cx = y, \tag{7}$$

which suggests that $\text{im}(C) \subseteq \text{im}(\Phi)$. It is easy to see that $\text{im}(\Phi) = \text{im}(CL^+C^T) \subseteq \text{im}(C)$. We can conclude that $\text{im}(C) = \text{im}(\Phi)$.

Since $L$ is a positive semidefinite matrix, we derive

$$x^T Lx = x^T C^T Cx = \|Cx\|_2^2, \tag{8}$$

which implies that $\ker(L) = \ker(C)$ and $\text{im}(C) = \text{im}(L)$. Given that $G$ is connected, $\dim(L) = \dim(C) = n - 1$. Combining this with $\text{im}(C) = \text{im}(\Phi)$, we have $\dim(\Phi) = \dim(L) = n - 1$, so the multiplicity of 1 is $n - 1$. Hence, we know the norm $\|\mathbb{E}yy^T\|_2 = \|\Phi\|_2 = 1$, $\text{tr}(\Phi) = n - 1$, and $\sum_e R_e = \text{tr}(\Phi) = n - 1$, which gives

$$p_e = \frac{(1 + S_e)R_e}{\sum_{e'}(1 + S_{e'})R_{e'}} \leq \frac{R_e}{\sum_{e'} 2R_{e'}} = \frac{R_e}{2(n-1)}. \tag{9}$$

We also have a bound on the norm of $y$: $\|\frac{1}{\sqrt{p_e}}\Phi_{:,e}\|_2 \leq \frac{1}{\sqrt{p_e}}\sqrt{\Phi_{e,e}} = \sqrt{2(n-1)}$. Now we are ready to apply Lemma D.1. By setting $q = 16c^2 n \log n/\epsilon^2$, we arrive at

$$\mathbb{E}\|\Phi S\Phi - \Phi^2\| = \mathbb{E}\|\frac{1}{q}\sum_{i=1}^q y_i y_i^T - \mathbb{E}yy^T\| \leq c\epsilon\sqrt{\frac{\log(16c^2 n \log n/\epsilon^2)2(n-1)}{16c^2 n \log n}} \leq \frac{\sqrt{2}\epsilon}{4}. \tag{10}$$

Using Markov's inequality, we get

$$\|\Phi S\Phi - \Phi^2\| \leq \epsilon \tag{11}$$

with high probability. Hence, with high probability, we have $\|\Phi S\Phi - \Phi^2\| \leq \epsilon$. This leads to the following equivalence:

Since $\|\Phi S\Phi - \Phi^2\| \leq \epsilon$, that is to say

$$\sup_{\mathbf{y}\in\mathbb{R}^m, \mathbf{y}\neq\mathbf{0}} \frac{|\mathbf{y}^T\Phi(S - I)\Phi\mathbf{y}|}{\mathbf{y}^T\mathbf{y}} \leq \epsilon.$$

Now, consider the case $\forall \mathbf{x} \in \mathbb{R}^n \setminus \ker(\mathbf{C})$, so $\mathbf{C}\mathbf{x} \in \mathbb{R}^m$ and $\mathbf{C}\mathbf{x} \neq 0$. We have

$$\sup_{\mathbf{x}\in\mathbb{R}^n, \mathbf{x}\notin\ker(\mathbf{C})} \frac{|\mathbf{x}^T\mathbf{C}^T\Phi(S - I)\Phi\mathbf{C}\mathbf{x}|}{\mathbf{x}^T\mathbf{C}^T\mathbf{C}\mathbf{x}} \leq \epsilon.$$

Recall that $\forall \mathbf{y} \in \text{im}(\mathbf{C})$, $\Phi\mathbf{y} = \mathbf{y}$ (Eq. (7)), thus we can substitute $\Phi\mathbf{C}\mathbf{x}$ with $\mathbf{C}\mathbf{x}$, leading to

$$\sup_{\mathbf{x}\in\mathbb{R}^n, \mathbf{x}\notin\ker(\mathbf{C})} \frac{|\mathbf{x}^T\mathbf{C}^T(S - I)\mathbf{C}\mathbf{x}|}{\mathbf{x}^T\mathbf{C}^T\mathbf{C}\mathbf{x}} = \sup_{\mathbf{x}\in\mathbb{R}^n, \mathbf{x}\notin\ker(\mathbf{C})} \frac{|\mathbf{x}^T\tilde{L}\mathbf{x} - \mathbf{x}^T L\mathbf{x}|}{\mathbf{x}^T L\mathbf{x}} \leq \epsilon.$$

Rearranging this inequality yields the desired result for $\mathbf{x} \notin \ker(\mathbf{C})$. For $\mathbf{x} \in \ker(\mathbf{C})$, it is trivial, since $\mathbf{x}^T L\mathbf{x} = \mathbf{x}^T \tilde{L}\mathbf{x} = 0$.

By the equivalence between $\|\Phi S\Phi - \Phi^2\| \leq \epsilon$ and $\frac{|\mathbf{x}^T\tilde{L}\mathbf{x} - \mathbf{x}^T L\mathbf{x}|}{\mathbf{x}^T L\mathbf{x}} \leq \epsilon$, and using the inequality

$$\mathbb{P}\left((1 - \epsilon)\mathbf{x}^T L\mathbf{x} \leq \mathbf{x}^T \tilde{L}\mathbf{x} \leq (1 + \epsilon)\mathbf{x}^T L\mathbf{x}\right) \geq 1 - \frac{\sqrt{2}}{4},$$

for $q = 16c^2 n \log n/\epsilon^2$, we conclude the proof. $\qquad\square$

$\hfill\square$

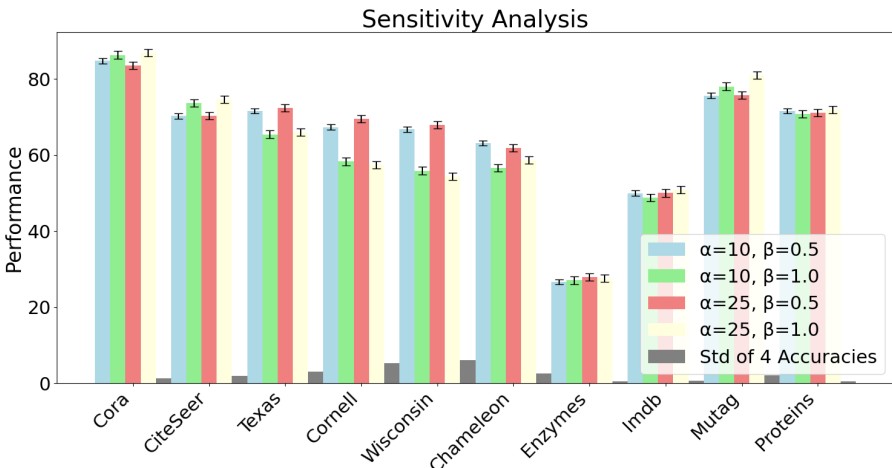

Figure 4: Hyperparameter sensitivity analysis on real-world datasets.

## E. Determining $|E_l|$

Note that $\kappa = \frac{\|Cx\|_\infty^2}{p_{\min}\|Cx\|_2^2}$, where $\|Cx\|_2^2 = x^T L x$ depends on the choice of $x$. Here, we select the leading eigenvector of $G$ to calculate $\kappa$, as our goal is to improve connectivity in the latent graph $G_l$, which requires significantly boosting the small eigenvalues. Using an eigenvector associated with the small eigenvalues of $G$ might make the small eigenvalues of $G$ and $G_l$ too similar, resulting in limited improvement in connectivity.

After calculating $\kappa$ and $q = \frac{\kappa^2}{2\epsilon^2} \log 8$, we estimate $|E_l|$ by solving the following problem. Suppose we apply sparsification on latent graph $G_l$ by sampling $q$ random edges with replacement, and observe $|E|$ distinct edges in the sparsified graph. The question is: what is the expected value of $|E_l|$, the number of edges in the latent graph? To simplify this computation, we assume a uniform edge probability distribution. Specifically, let $x$ denote the expected value of $|E_l|$. Then, we have

$$|E| = x \left(1 - \left(1 - \frac{1}{x}\right)^q\right). \tag{12}$$

Solving this equation yields $k = x - |E|$.

## F. Additional Experimental Results

### F.1. Hyperparameter Sensitivity Analysis

Table 9 presents the results of a node classification task for a sensitivity analysis of the hyperparameters $\alpha$ and $\beta$. For each node count (20, 200, and 2000 nodes) and homophily level (high and low), classification accuracy is evaluated across various combinations of $\alpha$ and $\beta$. The analysis shows that the classification accuracy is **relatively stable and not highly sensitive** to variations in these hyperparameters:

- **Alpha ($\alpha$)**:
  - For **smaller graphs** (e.g., 20 nodes), larger values of $\alpha$ $(0.15|E|)$ are generally preferred, as more aggressive densification appears to improve classification performance in smaller networks.
  - For **larger graphs** (e.g., 200 and 2000 nodes), the classification accuracy remains consistent across $\alpha$ values, with a slight tendency for smaller $\alpha$ $(0.05|E|)$ to perform better. This suggests that less aggressive densification is sufficient for improving the connectivity of larger networks.

- **Beta ($\beta$)**:
  - For **high homophily**, larger values of $\beta$ (e.g., 1.0) are generally preferred, as they help preserve the community structure by retaining more edges during sparsification.
  - For **low homophily**, smaller values of $\beta$ (e.g., 0.5) tend to perform better, as sparser graphs better reduce noise introduced by inter-community edges.

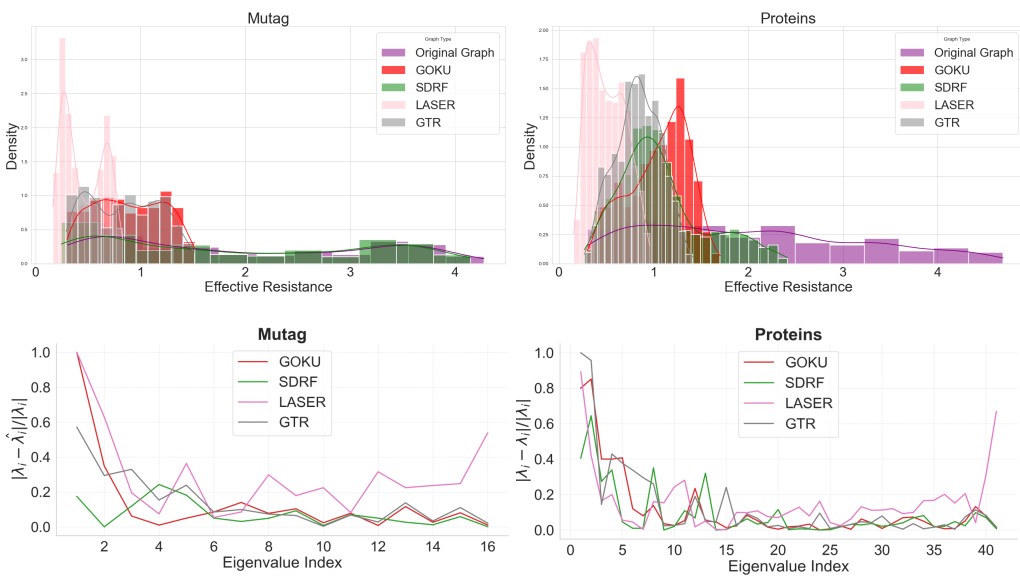

Figure 5: Trade-off between preserving spectrum and reducing ER.

We also present hyperparameter sensitivity analysis results on real-world datasets in Figure 4. Overall, the results demonstrate that **classification accuracy is not very sensitive** to changes in $\alpha$ and $\beta$ values within reasonable ranges. This suggests that GOKU's performance remains stable across a wide variety of hyperparameter settings, and the method is robust to changes in densification and sparsification levels.

### F.2. More Visualization Results

In Figure 6, we present additional visualizations of the spectral distributions for randomly selected graphs from the Mutag, IMDB, and Proteins datasets. Across all datasets, a consistent pattern emerges: GOKU demonstrates the highest fidelity in preserving the original graph spectra. This result underscores its effectiveness in striking a balance between enhancing connectivity and maintaining the structural integrity of the graph. By preserving spectral properties more effectively than alternative methods, GOKU ensures that key graph characteristics remain intact while improving overall connectivity.

Additionally, we provide Figure 5 for the trade-off between preserving spectrum and reducing ER.

| Nodes - Homophily | Hyperparameters ($\alpha$, $\beta$) | Classification Accuracy |
|---|---|---|
| **20 - High** | $(0.15\lvert E\rvert, 1.0)$ | 0.8765 |
| | $(0.15\lvert E\rvert, 0.7)$ | 0.8617 |
| | $(0.15\lvert E\rvert, 0.5)$ | 0.8541 |
| | $(0.1\lvert E\rvert, 1.0)$ | 0.8541 |
| | $(0.1\lvert E\rvert, 0.7)$ | 0.8392 |
| | $(0.1\lvert E\rvert, 0.5)$ | 0.8112 |
| | $(0.05\lvert E\rvert, 1.0)$ | 0.8321 |
| | $(0.05\lvert E\rvert, 0.7)$ | 0.8295 |
| | $(0.05\lvert E\rvert, 0.5)$ | 0.8189 |
| **20 - Low** | $(0.15\lvert E\rvert, 1.0)$ | 0.5734 |
| | $(0.15\lvert E\rvert, 0.7)$ | 0.5630 |
| | $(0.15\lvert E\rvert, 0.5)$ | 0.6334 |
| | $(0.1\lvert E\rvert, 1.0)$ | 0.5134 |
| | $(0.1\lvert E\rvert, 0.7)$ | 0.5231 |
| | $(0.1\lvert E\rvert, 0.5)$ | 0.5975 |
| | $(0.05\lvert E\rvert, 1.0)$ | 0.4901 |
| | $(0.05\lvert E\rvert, 0.7)$ | 0.5156 |
| | $(0.05\lvert E\rvert, 0.5)$ | 0.5967 |
| **200 - High** | $(0.15\lvert E\rvert, 1.0)$ | 0.9173 |
| | $(0.15\lvert E\rvert, 0.7)$ | 0.8851 |
| | $(0.15\lvert E\rvert, 0.5)$ | 0.9540 |
| | $(0.1\lvert E\rvert, 1.0)$ | 0.9540 |
| | $(0.1\lvert E\rvert, 0.7)$ | 0.9545 |
| | $(0.1\lvert E\rvert, 0.5)$ | 0.9632 |
| | $(0.05\lvert E\rvert, 1.0)$ | 0.8625 |
| | $(0.05\lvert E\rvert, 0.7)$ | 0.8321 |
| | $(0.05\lvert E\rvert, 0.5)$ | 0.7947 |
| **200 - Low** | $(0.15\lvert E\rvert, 1.0)$ | 0.5850 |
| | $(0.15\lvert E\rvert, 0.7)$ | 0.5661 |
| | $(0.15\lvert E\rvert, 0.5)$ | 0.6153 |
| | $(0.1\lvert E\rvert, 1.0)$ | 0.5534 |
| | $(0.1\lvert E\rvert, 0.7)$ | 0.5513 |
| | $(0.1\lvert E\rvert, 0.5)$ | 0.6012 |
| | $(0.05\lvert E\rvert, 1.0)$ | 0.4715 |
| | $(0.05\lvert E\rvert, 0.7)$ | 0.4898 |
| | $(0.05\lvert E\rvert, 0.5)$ | 0.5973 |
| **2000 - High** | $(0.15\lvert E\rvert, 1.0)$ | 0.9207 |
| | $(0.15\lvert E\rvert, 0.7)$ | 0.8918 |
| | $(0.15\lvert E\rvert, 0.5)$ | 0.8781 |
| | $(0.1\lvert E\rvert, 1.0)$ | 0.9127 |
| | $(0.1\lvert E\rvert, 0.7)$ | 0.8936 |
| | $(0.1\lvert E\rvert, 0.5)$ | 0.8625 |
| | $(0.05\lvert E\rvert, 1.0)$ | 0.9202 |
| | $(0.05\lvert E\rvert, 0.7)$ | 0.9282 |
| | $(0.05\lvert E\rvert, 0.5)$ | 0.8987 |
| **2000 - Low** | $(0.15\lvert E\rvert, 1.0)$ | 0.5335 |
| | $(0.15\lvert E\rvert, 0.7)$ | 0.5701 |
| | $(0.15\lvert E\rvert, 0.5)$ | 0.5934 |
| | $(0.1\lvert E\rvert, 1.0)$ | 0.5543 |
| | $(0.1\lvert E\rvert, 0.7)$ | 0.5921 |
| | $(0.1\lvert E\rvert, 0.5)$ | 0.5792 |
| | $(0.05\lvert E\rvert, 1.0)$ | 0.5213 |
| | $(0.05\lvert E\rvert, 0.7)$ | 0.5462 |
| | $(0.05\lvert E\rvert, 0.5)$ | 0.5814 |

Table 9: Node classification results for sensitivity analysis on hyperparameters ($\alpha = 0.05\lvert E\rvert, 0.1\lvert E\rvert, 0.15\lvert E\rvert$ and $\beta = 0.5, 0.7, 1.0$).

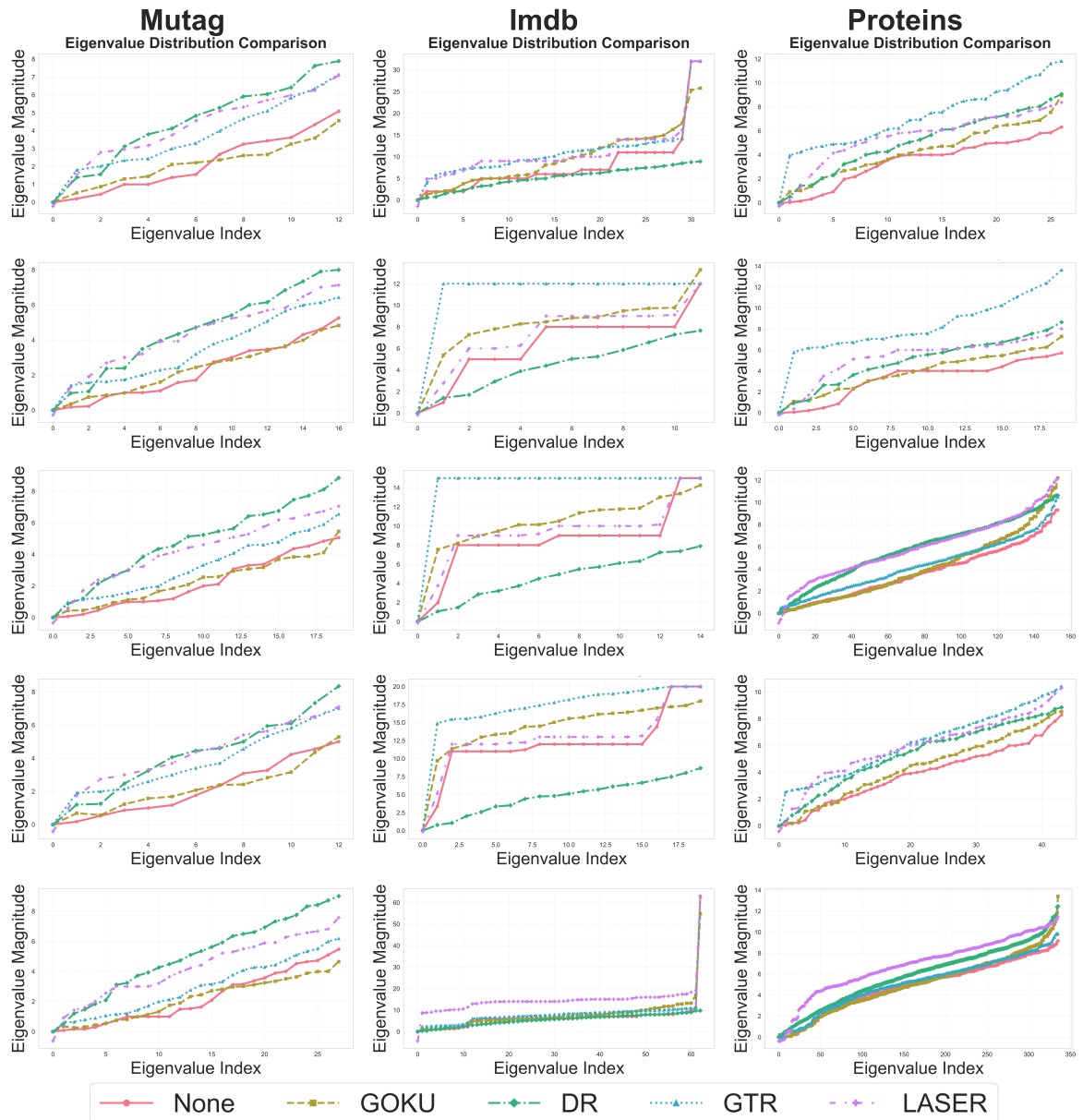

Figure 6: More randomly selected graph spectra visualization results from Mutag, Imdb, and Proteins datasets.

