# OpenReview forum: "Mitigating Over-Squashing in Graph Neural Networks by Spectrum-Preserving Sparsification"
_ICML.cc/2025/Conference — ICML 2025 poster_

### Official Review · Reviewer_EEAT · 2025-03-12

**Overall Recommendation:** 3

**Summary:**

This papers proposes a method a rewiring technique for graph neural networks called GOKU. The goal of GOKU is to improve the connectivity of the graph (as measured by the effective resistance) while maintaining the structure of the graph (as measured by the Laplacian spectrum). GOKU takes a two step approach: first, it uses a heuristic to reverse a sparsification process that is more likely to remove edges important for connectivity (in other words, this reverse sparsification will add edges back to the graph that improve connectivity). Then, it performs a variant of spectral sparsification on this densified graph that also aims to keep edges between vertices with similar features. The authors empirically test the ability of GOKU to preserve the Laplacian spectrum while decreasing effective resistance. Finally, the authors test compare GOKU to other rewiring algorithms as a pre-procesing step for graph neural networks.

**Claims And Evidence:**

The authors claim that GOKU is able to preserve the spectrum of the Laplacian. They argue this should be the case by constructing GOKU as a heuristic similar to an exact (but intractable) algorithm that would preserve the spectrum of the Laplacian. They demonstrate empirically comparing the spectrum of input graphs to those sparsified by GOKU. They also claim that GOKU will decrease the total resistance of nodes in the graph, which they demonstrate empirically. However, I am dubious of some of these claims; see Weaknesses. The authors claim that GOKU is an effective pre-processing step for graph neural networks, which they empirically verify by comparing it to other rewiring algorithms.

**Essential References Not Discussed:**

The paper makes a point that GOKU better preserves the spectrum of the graph than methods like FOSR, GTR, or LASER. However, I feel this somewhat misrepresents these methods. These papers recommend using these rewiring methods in conjunction with a relational GNN, which treats edges originally in the graph differently than those added during rewiring. (This could be true of the other rewiring methods as well, I am just most familiar with FOSR, GTR, and LASER.) Therefore, while I do think you can say that GOKU better preserves the structure of the graph than either of these methods, I think you should also mention that these methods aim to preserve the structure of the graph in a different way. (That said, you could argue that an advantage of using GOKU over these models is that you *don't* have to use a relational-GNN, which is a more complicated model with more parameters.)

**Experimental Designs Or Analyses:**

Mentioned above.

**Methods And Evaluation Criteria:**

The authors propose GOKU as a pre-processing step for a graph neural network. They compare GOKU to other rewiring algorithms and show that GOKU achieved SOTA performance compared to other rewiring algorithms on a set of standard GNN benchmarks.

**Other Comments Or Suggestions:**

Table 3 should say which model is being used. I also suggesting adding a None row to Table 3. (I know it is technically listed in Table 1, but it would make it easier on the reader to include it in Table 3 as well.)

I would recommend using the package thm-restate (or something similar) to copy the theorem statements of Theorem 4.1 and 4.2 to the appendix. This saves the reader from having to flip between the main text and the appendix when reading the proof of a theorem.

**Other Strengths And Weaknesses:**

# Strengths

The authors experimentally validate their claims that GOKU can preserve the Laplacian spectrum while decreasing effective resistance.

The authors have demonstrated the GOKU achieves SOTA performance as a pre-processing step for graph neural networks.

# Weaknesses

There are a few inconsistencies in how this paper is presented in the introduction (see the Questions section below). Namely, the introduction presents the algorithm as both maintaining graph structure (in terms of the Laplacian spectrum) while improving connectivity (in terms of effective resistance), which is a contradiction as the effective resistances are a function of the Laplacian spectrum. (For example, the sum of effective resistance between all pairs of nodes is the trace of the pseudoinverse of the Laplacian; see e.g., (Ghost et al, 2006, Equation 15).)

Ghosh, A., Boyd, S., & Saberi, A. (2008). Minimizing effective resistance of a graph. SIAM review, 50(1), 37-66.

**Questions For Authors:**

Would it be possible to do an ablation similar to Table 3 where you include/exclude the term $S_e$ from the probabilities in ISS step? Right now, it is not clear to me how much of the improvement in GOKU is due to preserving the spectrum vs. favoring edges with similar features. This feels relevant as the next best method is Delauney Rewiring, which as you note does not account for graph topology at all and only accounts for similarity of features.

You claim GOKU is both a spectral sparsifier while also reducing the effective resistance between most nodes. However, this seems to be a contradiction, as spectral sparsifiers preserve effective resistance. How is GOKU able to both preserve the spectrum of the graph while also reducing effective resistance? To this point, you show the effective resistance distribution (Figure 3) and eigenvalues of graphs (Figure 2) where GOKU was applied, but never for the same graph. Could you include both the effective resistance distribution and eigenvalues of a single graph so we could see the trade-off between these two?

The introduction of your paper says that the goal is to increase the smallest eigenvalues of the graph Laplacian while keeping the rest fixed. However, your theoretical results and your experiments both suggest that you preserve all eigenvalues. Why do you say you increase the smallest eigenvalues?

**Relation To Broader Scientific Literature:**

This paper fits into a line of other works aimed at tackling over-squashing in graph neural networks by changing the topology of the graph. The technique in this paper seems most inspired by the work of (Karhadkar et al 2023) that aims to address oversquashing by increasing the spectral gap of the Laplacian, although the use of effective resistance as a measure for connectivity was proposed by (Black et al 2023, Di Giovanni et al 2023, 2024) who theoretically showed that over-squashing occurs between nodes of high effective resistance. In contrast to previous rewiring techniques, GOKU also aims to preserve the structure of the graph (as measured by the Laplacian spectrum), whereas other techniques focus on changing the graph to improve its connectivity. To maintain the spectrum of the graph, this paper relies on spectral sparsification techniques (Spielman and Srivistava, 2011).

**Theoretical Claims:**

The authors prove that the two sparsification algorithms they consider are provably spectral sparsifiers. However, there are no guarantees for the unsparsification step, instead using on a heuristic for this step. Therefore, the entire algorithm GOKU has no theoretical guarantees of spectral sparsification.

---

> ### Author Rebuttal · Authors · 2025-03-31
>
> We sincerely thank the reviewer for their time, effort, and constructive comments. Below are our responses.
>
> > ### **Comment 1**: Preserve all eigenvalues and the smallest eigenvalues are not increased in theoretical & experimental results
>
> Our method indeed increases the smallest eigenvalues. Here is a more detailed explanation:
>
> 1. **Theoretical Results.** In Theorem 4.1, we show that $(1 - \\epsilon) x^T L x \\leq x^T \\tilde{L} x \\leq (1 + \\epsilon) x^T L x$ holds conditional on $\\kappa = \\kappa(x)$, **a function of the eigenvector $x$**. Specifically, we choose the leading eigenvector $x_l$ (with the largest eigenvalue), and derive the sampling count $q \\ge \\frac{\\kappa(x_l)^2}{2\\epsilon^2} \\log 8$. This derivation focuses on preserving the larger eigenvalues and does not account for the smallest eigenvalues.
>
> 2. **Experimental Results.** Our experiments do show that the smallest eigenvalues are increased. However, due to the **large range** of eigenvalues, particularly the largest eigenvalue, observing this increment directly can be challenging. For instance, in Figure 2 (right), the spectral gap increases from nearly 0 to 0.5. Given that the largest eigenvalue is around 14, this makes the smallest eigenvalue changes **difficult to observe**. To make this clearer, we will include a zoom-in on the smallest eigenvalues for better visualization.
>
> ---
>
> > ### **Comment 2:** How to reduce Effective Resistance (ER) While Preserving Eigenvalues
>
> Note that the total ER, $R_{\\text{tot}} = n \\sum_{i=2}^n \\frac{1}{\\lambda_i}$, is **sensitive to smallest eigenvalues**. For example, as shown in Figure 2 (right), the largest eigenvalue is approximately 14, contributing only $\\frac{1}{14}$ to $R_{\\text{tot}}$. On the other hand, the second smallest eigenvalue $\\lambda_2 = 0.05$ contributes significantly more ($\\frac{1}{\\lambda_2} = 20$). By increasing the smallest eigenvalue, for instance, to $\\lambda_2 = 0.5$, its contribution reduces to 2. Hence, by slightly increasing a few of the smallest eigenvalues, we can significantly reduce $R_{\\text{tot}}$, while maintaining a similar spectrum for others.
>
> See visualization for this trade-off on a single graph: https://anonymous.4open.science/r/Submission3560_Rebuttal_for_Reviewer_EEAT/Fig.pdf
>
> ---
>
> > ### **Comment 3:** Ablation Study on $S_e$
>
> We provide the following clarifications for the impact of $S_e$:
>
> - **Impact of $S _e$:** The sampling probability is determined by $(1 + S_e)R_e$, where $S_e \\in [0,1]$ adds only a bias term to the factor 1, and $R _e$ remains the dominant factor. For example, consider two edges $a$ and $b$ with $S_a = 1$, $R_a = 0.5$ and $S _b = 0.5$, $R_b = 1$. We get: $(1+S_a)R_a = 2 \\times 0.5 = 1$ and $(1+S _b)R_b = 1.5 \\times 1 = 1.5$. Here, **$R_e$ plays a more significant role than $S_e$**. This is why our focus is on preserving the spectrum.
>
> - **Ablation Results:** We conduct an ablation study to analyze the role of $S_e$ in graph-level and node classification tasks across both homophily and heterophily datasets. We find that the utilization of **$S_e$ contributes less to performance** in graph-level and homophily tasks. On the other hand, Delaunay Rewiring, which disregards the original graph topology, still performs well in certain datasets. This is due to Delaunay triangulation’s inherent properties that ensure the graph remains well-connected, which is crucial for graph-level tasks.
>
> Here is the ablation study:
>
> |  | Mutag | Imdb  | Proteins | Chameleon | Cora  |
> |----|----|----|---|-----|---|
> | GOKU-Non$S_e$| 80.4  | **50.2** | 71.4     | 59.6      | 86.1  |
> | GOKU   | **81.0** | 49.8  | **71.9** | **63.2**  | **86.8** |
>
> ---
>
> > ### **Comment 4**: Writing suggestions on adding a None row to Table 3, restating the theorems & Relational-GNN
>
> We thank the reviewer for the valuable suggestions and feedback!
>
> 1. We will explicitly mention that model used in Table 3 is the GCN models from Table 1. We also agree with adding a "None" row for clarity and restating the theorem statements in the appendix, which we believe will make it more reader-friendly. We will follow this suggestion in our future works as well. Thank you!
> 2. We will clarify that methods such as FOSR, GTR, or LASER leverage Relational-GNN to preserve graph structure, whereas our method has the advantage that it doesn’t require the use of Relational-GNN to achieve this. This distinction will help readers better understand the literature and our contributions.
>
> > ### **Comment 5**: Theoretical guarantees for entire algorithm
>
> We acknowledge that while the sparsification algorithm provides provable spectral sparsifiers, the reverse process (densification) does not have the same theoretical guarantees. It performs well empirically. The number of edges added during densification is **derived using Theorem 4.1 and the leading eigenvector** by $q \\ge \\frac{\\kappa(x_l)^2}{2\\epsilon^2} \\log 8$, rather than being chosen heuristically.

---

> > ### Comment · Reviewer_EEAT · 2025-04-01
> >
> > Thank you for your response! I see now the distinction between preserving the spectrum and reducing total resistance, namely, that you are mostly reducing the spectrum of the small eigenvalues, which can be hard to see on plots of the eigenvalues (which your new plots address.) As you mention, even increasing the small eigenvalues a little can have a large increase on the total resistance, which was the detail I did not understand before. I also appreciate the new figure comparing the resistance and spectrum on the same graph and the ablation study on $S_e$. I believe all of your answers help clarify the paper.

---

> > > ### Author Response · Authors · 2025-04-02
> > >
> > > Dear Reviewer EEAT,
> > >
> > > Thank you for your prompt reply! We are pleased to hear that our responses have helped address your concerns. Once again, we sincerely appreciate your valuable feedback and suggestions. We wish you the best of luck with your own submission if any. Thank you!
> > >
> > > Best regards,
> > >
> > > The Authors

---

### Official Review · Reviewer_yvpT · 2025-03-13

**Overall Recommendation:** 4

**Summary:**

This paper addresses the issue of over-squashing in Graph Neural Networks (GNNs) by introducing a novel approach called the **Densification-Sparsification Rewiring** framework, with a practical implementation termed **GOKU**. The main contribution of the paper is the proposal of a two-step rewiring process:

1. Densification: This step applies an **unimportance-based sparsification** algorithm to solve an inverse spectral graph sparsification problem, aiming to improve graph connectivity.
2. Sparsification: The second step uses an **importance-based sparsification** algorithm, ensuring the graph remains sparse but retains crucial connectivity.

The combination of these two steps allows GOKU to preserve both the graph’s spectral properties and density while improving overall connectivity, which helps mitigate the problem of over-squashing in GNNs.

In addition to the framework, the authors provide theoretical results that guarantee spectral preservation and include a time complexity analysis to demonstrate the efficiency of GOKU. The paper supports these contributions with extensive experiments on node and graph classification tasks, ablation studies, and visualizations of graph spectra. These experiments show that GOKU is a promising solution for better graph rewiring in GNNs.

**Claims And Evidence:**

The claims are supported by experimental results and thorough analyses.

**Essential References Not Discussed:**

Related works are adequately discussed in the paper, including literature on graph rewiring, graph scarification, and other relevant graph neural network studies.

**Experimental Designs Or Analyses:**

The paper evaluates the method from multiple perspectives, including performance, rewiring effects on graph structure, and efficiency, using both homophilic and heterophilic node classification datasets, as well as graph classification datasets. Especially, the classification experiments and spectral visualizations effectively demonstrate that the proposed method meets its goals of improving connectivity and preserving graph spectra.

**Methods And Evaluation Criteria:**

The proposed method appears effective in addressing the over-squashing problem while maintaining spectrum similarity between the original and rewired graphs. The authors used various datasets including on both node and graph classification to comprehensively evaluate the proposed method.I have roughly reviewed the proofs of the main theorems, and they appear to be sound.

**Other Comments Or Suggestions:**

N/A.

**Other Strengths And Weaknesses:**

***Strengths***

- The originality of the method lies in its ability to integrate spectral preservation into the graph rewiring process, along with a novel design of inverse graph sparsification, which is both interesting and promising.
- The experimental evaluation is comprehensive, utilizing both homophilic and heterophilic node classification datasets, as well as graph classification datasets, to thoroughly assess the method’s performance.
- The paper is well-written and easy to follow.
- Extensive experimental results validate the effectiveness of the proposed method in improving connectivity and preserving spectral properties.


***Weaknesses***

- The authors use two specific spectral sparsification algorithms (Algorithm 1 and Algorithm 2), but the reasons for selecting these particular algorithms are not sufficiently discussed. It would be helpful for the authors to explain why these algorithms were chosen over others. Can these algorithms be replaced with alternative spectral sparsification methods?

- The GOKU method relies on random sampling, but the potential impact of this randomness on the quality and properties of the rewired graphs is not fully explored. A more detailed analysis of how random sampling may affect the consistency and reliability of the rewired graphs would strengthen the paper's conclusions.

**Questions For Authors:**

See weaknesses part.

**Relation To Broader Scientific Literature:**

The proposed method offers a potential new paradigm for graph rewiring techniques, specifically addressing the over-squashing issue while preserving the spectral properties of the original graphs. This contribution builds on prior work by integrating spectral considerations into graph rewiring, an aspect that has not been fully explored in previous methods.

**Theoretical Claims:**

I have roughly reviewed the proofs of the main theorems, and they appear to be sound.

---

> ### Author Rebuttal · Authors · 2025-04-01
>
> We sincerely thank the reviewer for their time, effort, and constructive comments. Here are our responses.
>
> ### **Comment 1**: Choices of Spectral Sparsification Algorithms
>
> We appreciate the reviewer’s interest in the choice of spectral sparsification algorithms. It is important to note that our approach is designed to be algorithm-agnostic, meaning it does not rely on any particular sparsification method. However, we acknowledge that different algorithms may offer distinct advantages, and we discuss two such algorithms here as representative examples. Other similar algorithms can also be considered within our framework.
>
> 1. **Algorithm 1**: Algorithm 1 is designed to selectively increase the smaller eigenvalues while maintaining the larger ones. This is particularly important when the goal is to approximate the spectral properties of the graph. In **Theorem 4.1**, we show that for any vector $x$, the following inequality holds:
>    $$
>    (1 - \\epsilon) x^T L x \\leq x^T \\tilde{L} x \\leq (1 + \\epsilon) x^T L x
>    $$
>    This result is conditional on $\\kappa = \\kappa(x)$, which is a function of the eigenvector $x$. Specifically, we choose the leading eigenvector $x_l$ (corresponding to the largest eigenvalue) and derive the minimum required sampling count, $q \\geq \\frac{\\kappa(x_l)^2}{2\\epsilon^2} \\log 8$. This derivation focuses on ensuring that the larger eigenvalues are preserved, while the smaller eigenvalues are not explicitly accounted for in this process.
>
> 2. **Algorithm 2**: This algorithm assigns higher probabilities to edges with larger effective resistance, which is aligned with our objective of preserving important edges during the sparsification process. We believe this approach is well-suited to maintain key structural properties of the graph. The algorithm presented in [1] is one such effective candidate for implementing Algorithm 2, but we note that other algorithms with similar principles could also be used within the framework of our method.
>
> ---
>
> ### **Comment 2**: Randomness in Sampling
>
>
> Our theorems explicitly account for the randomness in the sampling process by providing probability lower bounds for the properties of the sampled graphs. Specifically, once the sampling count $q$ exceeds a certain threshold, we show that the generated graphs exhibit the desired spectral properties with high probability. These theoretical results ensure that, despite the inherent randomness in the sampling, the method remains reliable and effective.
>
> To further substantiate the reliability of our approach, we conducted extensive empirical validation. As shown in **Tables 1 and 2**, our results are based on the average of 100 independent runs with different random seeds. This approach allows us to assess the variability and consistency of our algorithm. The results demonstrate that the confidence intervals of our algorithm are comparable to those of deterministic methods, which reinforces the robustness of our approach despite the use of random sampling.
>
> We believe that these empirical and theoretical analyses collectively validate the effectiveness of our algorithm, providing confidence in its consistency and reliability.
>
> ---
>
> We again thank the reviewer for the constructive feedback and valuable suggestions. Please do not hesitate to reach out if any further clarifications are needed.
>
> ---
> ### References
> [1] Spielman, Daniel A., and Shang-Hua Teng. "Spectral sparsification of graphs." SIAM Journal on Computing 40.4 (2011): 981-1025.

---

### Official Review · Reviewer_geiv · 2025-03-14

**Overall Recommendation:** 2

**Summary:**

This paper introduces GOKU, a novel graph rewiring method that mitigates over-squashing in Graph Neural Networks through a two-phase densification-sparsification paradigm. The approach first identifies and adds critical missing edges to alleviate bottlenecks via inverse spectral sparsification, then selectively removes less important edges to maintain sparsity while preserving spectral properties. The results demonstrate that GOKU outperforms existing rewiring methods on multiple node and graph classification tasks.

**Claims And Evidence:**

1. The paper claims that GOKU has "nearly linear time" with respect to the number of nodes and edges. However, it lacks a thorough explanation of how the complexity of approximate Effective Resistance calculations affects the overall complexity.

2. While the paper emphasizes spectrum preservation, the actual implementation appears to prioritize feature similarity in the sparsification process. This suggests the possibility that it optimizes other objectives (e.g., improved homophily) rather than purely preserving the spectrum.

3. Although the paper emphasizes theoretical guarantees for spectrum preservation, the visualization of spectrum comparisons in Section F.2 provides only qualitative assessment without quantitative measurements (e.g., spectral similarity metrics).

**Essential References Not Discussed:**

PR-MPNN [1], which proposes a rewiring strategy that structures the graph according to downstream tasks and underlying data distribution, is missing. Studies that improve message passing methods without relying on rewiring are also omitted. For instance, there should be discussion of Co-GNN [2], which learns behavior when updating node features, or PANDA [3] and GESN [4], which aim to mitigate over-squashing according to model characteristics while avoiding rewiring methods.

> [1] Qian, Chendi, et al. "Probabilistically rewired message-passing neural networks." ICML 2024
>
> [2] Finkelshtein, Ben, et al. "Cooperative Graph Neural Networks." ICML 2024.
>
> [3] Choi, Jeongwhan, et al. "PANDA: Expanded Width-Aware Message Passing Beyond Rewiring." ICML 2024
>
> [4] Tortorella, Domenico, and Alessio Micheli. "Leave Graphs Alone: Addressing Over-Squashing without Rewiring." The First Learning on Graphs Conference.

**Experimental Designs Or Analyses:**

1. Reporting that hyperparameters are not sensitive without providing sensitivity studies for alpha and beta lacks persuasiveness.

2. The paper claims GOKU has "nearly linear time" with respect to nodes and edges, but doesn't explain in detail how the complexity of approximate Effective Resistance calculations affects the overall complexity. Particularly, can $\mathcal{O}(|E'|m)$ be considered linear complexity when $|E'|$ is large?

**Methods And Evaluation Criteria:**

The paper claims that GOKU outperforms existing methods on several datasets, which is well supported by extensive experimental results in Tables 1, 2, and 7.

**Other Comments Or Suggestions:**

1. The paper states that it aims to solve the over-squashing problem, but the experimental evaluation mainly focuses on classification accuracy. There is a lack of direct measurement of how much over-squashing has actually been reduced. More appropriate evidence would be measuring signal propagation to distant neighbors (as in [5] and [3]) or providing theoretical bounds on over-squashing sensitivity.

2. Could a more appropriate explanation be added for the ER change process from 1.5 → 0.3 → 0.5 in Fig 1? It would be helpful to better explain the intuition behind why it increases again from 0.3 to 0.5 after sparsification.

3. It would be beneficial if the authors could analyze which newly added edges and which of those added edges ultimately remain after sparsification.

> [5] Di Giovanni, Francesco, et al. "On over-squashing in message passing neural networks: The impact of width, depth, and topology." ICML 2023.
>
> [3] Choi, Jeongwhan, et al. "PANDA: Expanded Width-Aware Message Passing Beyond Rewiring." ICML 2024

**Other Strengths And Weaknesses:**

**Strengths**

1. This paper introduces a paradigm that aims to solve over-squashing, maintain graph sparsity, and preserve spectral properties.

2. The method proposed by the authors consistently demonstrates performance improvements, particularly on heterophilic graphs.

**Weaknesses**

1. GOKU has a disadvantage that, unlike other previous studies, sparsification and densitification must be performed separately.

2. Despite being the primary motivation, this paper lacks direct measurements of over-squashing and instead relies on classification accuracy.

3. The hyperparameter explanations for $\alpha$ and $\beta$ are inconsistent. Specifically, for $\alpha$, it is described on page 7 as $\alpha \in \{5, 10, 15, 20, 25, 30\}$, but in the Appendix it appears to use 0.05, 0.1, 0.15. For $\beta$, page 7 only considers 0.5 and 1.0, but the Appendix considers 0.5, 0.7, and 1.0.

4. The authors explain the process of adding edges during the densification phase and then deleting edges during the sparsification phase, but authors do not provide specific analysis on how many of the newly added edges are retained or deleted, or by what criteria. Moreover, even in the experimental section of the paper, there is no analysis of the survival rate of newly added edges or their role in the final graph.

**Questions For Authors:**

1. For $\alpha$, on page 7 it is described as $\alpha \in \{5, 10, 15, 20, 25, 30\}$, but in the Appendix it appears to use 0.05, 0.1, 0.15. For $\beta$, page 7 only considers 0.5 and 1.0, but the Appendix considers 0.5, 0.7, and 1.0. Could you clarify these hyperparameter specifications?

2. On page 16, you state that "Overall, the results demonstrate that classification accuracy is not very sensitive to changes in $\alpha$ and $\beta$ values within reasonable ranges." Could you report sensitivity studies on alpha and beta with finer granularity to confirm they are truly not sensitive?

3. On page 5, line 260, you state "The USS method should assign low probabilities $p_e$ to edges crucial for maintaining connectivity." Shouldn't important edges be assigned high probabilities? Looking at Algorithm 1, it seems designed to assign low probabilities to them.

4. Why is the range for $\beta$ set as $0.5 < \beta ≤ 1$? Is there a rationale for this specific range?

5. You mention that setting $δ = 0.1$ results in an error of $(1 ± (1 + δ)ϵ)$. Could you analyze how this approximation error affects the overall algorithm performance?

6. In Fig 1, the ER changes from 1.5 → 0.3 → 0.5. Why does it increase again from 0.3 to 0.5 after sparsification?

7. The example in Fig 1 shows that newly added edges can be removed again by sparsification. Does this happen in your actual experiments?

8. Could you analyze which newly added edges ultimately remain after the sparsification phase in your experiments?

9. Have you tried experimenting with reversing the order of densification and sparsification?

**Relation To Broader Scientific Literature:**

The DSR represents a conceptual shift in graph rewiring approaches:

- Unlike previous methods that either only add edges (Karhadkar et al., 2023) or only modify existing ones (Barbero et al., 2024), DSR employs a two-phase process. This connects to optimization literature, where intermediate relaxations are used before refinement to final solutions.
- The paper addresses the tension between connectivity enhancement and over-smoothing risk, similar to how ProxyGap (Jamadandi et al., 2024) leverages the Braess paradox to show that removing edges can sometimes improve connectivity. GOKU's approach aligns more with the recent LASER rewiring (Barbero et al., 2024), which also aims to maintain aspects of the original graph structure.

**Theoretical Claims:**

1. In Section 4.1, the authors acknowledge the difficulties of MLE calculation and propose a simplified approach, but there's a lack of theoretical guarantees regarding how accurately this simplified approach solves the original MLE problem.

2. Theorem 4.2 claims to guarantee spectral similarity with high probability, but doesn't present specific probability bounds (only expressed as "with high probability").

3. In the ISS algorithm, the theoretically proposed probability distribution is modified by additionally considering feature similarity. It appears that the impact of this modification on the guarantees in Theorem 4.2 is not analyzed.

---

> ### Author Rebuttal · Authors · 2025-03-31
>
> We sincerely thank the reviewer for their time, effort, and constructive comments. Below are our responses.
>
> > ### **Comment 1 (Claims And Evidence 2, Theoretical Claims 2&3)**: Lack of probability bounds, impact of feature similarity in Theorem 4.2 & feature similarity term in ISS (Algorithm2)
>
> - For $q= 32c^2n\\log n/\\epsilon^2$, the lower bound is $3/4$.
> - Theorem 4.2 already accounts for feature similarity (**Eq. 9** in Appendix).
> - We don't prioritize feature similarity. For instance, given edges $a$ and $b$ with $S_a=1,R_a=0.5$ and $S_b=0.5,R_b=1$, we have sampling probabilities proportional to $(1+1)*0.5=1$ for $a$ and $(1+0.5)*1=1.5$ for $b$.
>
> > ### **Comment 2 (Claims And Evidence 1 & Experimental Designs 2)**: Time complexity
> - We discuss the time complexity of approximating ER in Sec. 4.3 as $\\mathcal{O}(m \\cdot \\text{poly}(\\log n)/\\delta^2)$, where $\\text{poly}(\\log n)$ is a polynomial in $\\log n$.
> - If $|E'|$ is too large, $\\mathcal{O}(|E'| m)$ is not sublinear. However, this is not an issue in practice: in dense graphs, we don’t need large $|E'|$ to achieve good connectivity. In sparse graphs (small $m$), large $|E'|$ won’t significantly increase $|E'| m$. For Tables 1 and 2, we chose $|E'|$ from \{5, 10, 15, 20, 25, 30\}.
>
> > ### **Comment 3 (Weaknesses 2, Suggestions 1 & Claims And Evidence 3)**: Measurement of reduction in over-squashing & quantitative measurements of spectrum similarity
> * We measure the reduction in over-squashing by comparing **pairwise ER** between the original and rewired graphs in **Fig. 3**, which shows that ER values decrease post-rewiring.
> * Smaller ER $R_{u,v}$ leads to larger $\\left\\| \\frac{\\partial h_u^{(r)}}{\\partial x_v} \\right\\|$(Black, Mitchell, et al.) the same metric as in Di Giovanni et al. Smaller $R_{u,v}$ also implies more and shorter paths.
>
> See visualization: https://anonymous.4open.science/r/Submission3560_Rebuttal_for_Reviewer_geiv/Fig.pdf
>
> > ### **Comment 4**: Not discussed references
>
> We mainly focused on rewiring-based work in the submission. Thanks for providing these excellent references! We will discuss them in the revision.
>
> > ### **Q1**: $\\alpha$ and $\\beta$ search spaces
>
> Apologies for any confusion. The search spaces for $\\alpha$ and $\\beta$ in page 7 apply to Tables 1 and 2.
> - **$\\alpha$ in the Appendix**: Table 8 analyzes $\\alpha$ sensitivity, not GOKU vs. baselines. For the 2000-node, 80k-edge graphs, a small search space for $\\alpha$ (5 to 30) is too limited for capturing meaningful difference. We use $0.05|E|, 0.1|E|, 0.15|E|$ for this reason.
> - **$\\beta$ values**: For the main experiments, we use $\\{0.5, 0.6, 0.7, 0.8, 0.9, 1.0\\}$. For Table 8, we use $\\{0.5, 0.7,1.0\\}$ to avoid an overly large table while capturing key trends.
>
> > ### **Q2**: Sensitivity studies
>
> We summarize the average accuracy across $18 = 3 \\times 6$ settings in Table 8:
> - $\\alpha = (0.05|E|, 0.1|E|, 0.15|E|)$: 0.705, 0.728, 0.731
> - $\\beta = (0.5, 0.7, 1.0)$: 0.730, 0.717, 0.705
> The variances of the average accuracies are relatively small.
>
> > ### **Q3**: USS
>
> USS assigns **low probabilities to important edges**, often leading to their removal. However, we don't apply USS directly; instead, we reverse the process (densification) to reintroduce important edges, mitigating over-squashing.
>
> > ### **Q4**: Range of $\\beta$
>
> To ensure efficiency, we constrain $\\beta \\leq 1$ to avoid denser output graphs than input. We set $\\beta \\geq 0.5$ empirically to prevent overly sparse graphs and reduce the hyperparameter search space.
>
> > ### **Q5**: ER Approximation error $\\delta$
>
> The error bound $1 \\pm (1+(\\delta))\\epsilon$ can be derived by scaling $p_e$ as in Eq. 9 of the Appendix. Our experiments show minimal performance drops for $\\delta = 0.4$ compared to Table 1($\\delta=0.1$).
>
> ||Cora|Texas|Mutag|
> |-|-|-|-|
> | $\\delta=0.4$|↓0.3|↓0.5|↓1.2|
> | $\\delta=0.7$|↓0.6|↓1.2|↓2.6|
> | $\\delta=1.0$|↓0.4|↓2.8|↓2.4|
>
> > ### **Q6**: ER changes in Fig. 1
>
> Sparsification is not meant to further improve connectivity after densification. ER increases when edges are removed (Rayleigh Monotonicity). There is a trade-off: as discussed in **Page 8**, sparsification provides two benefits: reducing edge density and retaining spectral similarity. **Fig. 1** shows that sparsified graph has a more similar spectrum to the original graph.
>
> > ### **Q7 & Q8**: Newly added edges survival
>
> While sparsification may remove newly added edges, they are assigned higher sampling probabilities during sparsification as they are topologically important. We computed the survival rate using the three datasets mentioned above. Over 90% of these edges were retained after sparsification.
>
> > ### **Q9:** reversing the order of DSR
>
> We have not explored reversing the order of densification and sparsification. If we apply sparsification first, we have to densify a weighted graph, which is more diffcult. We leave this interesting question for future work.

---

> > ### Comment · Reviewer_geiv · 2025-04-04
> >
> > Thank you for addressing many of my questions. However, I still have some questions:
> >
> > Q1. The authors state that the lower bound is 3/4, but could you explain the relationship between $q \geq \frac{\kappa^2}{2\epsilon^2}\log 8$ used in Theorem 4.1 in the main paper and $q = 32c^2n\log n/\epsilon^2$ mentioned in the rebuttal?
> >
> > Q2. Regarding Figure 1 provided in the link: Why did you use a synthetic dataset instead of the datasets used in the paper, and could you provide comparison results with other baseline methods?
> >
> > Q3. Regarding Figure 2 provided in the link: How does the significant change in the smallest eigenvalues align with the goal of spectral preservation? Also, could you quantitatively present the eigenvalue preservation performance compared to other baseline methods?
> >
> > Q4. In Figure 1's caption, you state it is "providing a direct measure of over-squashing reduction," but shortest path distributions may not be directly related to over-squashing in GNNs. Could you explain this relationship with reference to or comparison with over-squashing measurement methods proposed in papers such as [1,2]?
> >
> > > [1] Di Giovanni, Francesco, et al. "On over-squashing in message passing neural networks: The impact of width, depth, and topology." ICML 2023.
> > >
> > > [2] Choi, Jeongwhan, et al. "PANDA: Expanded Width-Aware Message Passing Beyond Rewiring." ICML 2024
> >
> >
> > Q5. You haven't discussed the accuracy of your approach to simplifying the MLE problem. Could you explain how well your proposed MLE approach approximates the theoretical MLE problem?
> >
> > Q6. Could you provide detailed sensitivity analysis graphs showing how different values of $\alpha$ and $\beta$ affect performance for each dataset?
> >
> > Q7. To establish the design rationale of DSR, I think it's necessary to explore reversing the order of DSR. Could you specifically explain why densifying a weighted graph is more difficult?
> >
> > Q8. Are the assumptions related to Lemma D.1 used in the proof of Theorem 4.2 still valid after introducing feature similarity?
> >
> > Q9. Do you have evidence that the condition $\mathbb{E}[yy^\top] = \Phi$ used in the proof of Theorem 4.2 is still satisfied when including feature similarity in $p_e \propto (1 + S_e) R_e$? I ask because you mentioned this proof follows the framework of Spielman & Srivastava (2008).

---

> > > ### Author Response · Authors · 2025-04-07
> > >
> > > We sincerely thank the reviewer for their valuable feedback and insightful follow-up questions. Below are our responses:
> > >
> > > ### Q1.
> > >
> > > The expression $q \ge \frac{\kappa(x)^2}{2\epsilon^2}\log 8$ in Theorem 4.1 and $q = \frac{32c^2n\log n}{\epsilon^2}$ in Theorem 4.2 have distinct roles:
> > > - Theorem 4.1 is focused on **densification**, where we aim to amplify the smallest eigenvalues and can choose an eigenvector $x$ corresponding to large eigenvalues that we wish to preserve.
> > > - Theorem 4.2, used for **sparsification**, aims to preserve all eigenvalues while reducing the edge count. The expression $q = \frac{32c^2n\log n}{\epsilon^2}$ is independent of eigenvectors, making it more general and larger, as it considers all possible eigenvectors.
> > >
> > > In summary, $q = \frac{32c^2n\log n}{\epsilon^2}$ is typically larger because it accounts for all eigenvectors in $\mathbb{R}^n$.
> > >
> > > ### Q2.
> > >
> > > Please see the link below
> > >
> > > ### Q3.
> > >
> > > As stated in line 82, our goal is to slightly increase the smallest eigenvalues of the Laplacian while maintaining sparsity and preserving other eigenvalues. The total effective resistance (ER) is highly sensitive to these smallest eigenvalues, which are often close to zero. Even a small increase in these values (e.g., from 0.1 to 0.2) can result in a significant relative change in ER.
> > >
> > > ### Q4.
> > >
> > > The effective resistance (ER) in our paper is closely related to the Jacobian norm measurements found in [1, 2], as detailed in Theorem 3.3 of [3]:
> > >
> > > $$
> > > \left\| \frac{\partial h_u^{(r)}}{\partial x_v} \right\| \leq (2\alpha\beta)^r \frac{d_{\text{max}}}{2} \left( \frac{2}{d_{\text{min}}} \left( r + 1 + \frac{\mu^{r+1}}{1-\mu} \right) - R_{u,v} \right)
> > > $$
> > >
> > > where a smaller ER $R_{u,v}$ leads to a larger Jacobian norm $\left\| \frac{\partial h_u^{(r)}}{\partial x_v} \right\|$. Increasing the smallest eigenvalues reduces total ER, thereby increasing the **global** Jacobian norm, as expressed in Corollary 3.6 in [3]:
> > >
> > > $$
> > > \sum_{u \neq v \in V} \left\| \frac{\partial h_u^{(r)}}{\partial x_v} \right\| \leq (2\alpha \beta)^r \frac{d_{\text{max}}}{2} \left( n \cdot (n-1) \frac{1}{d_{\text{min}}} \left( r + 1 + \frac{\mu^{r+1}}{1-\mu} \right) - R_{\text{tot}} \right)
> > > $$
> > >
> > >
> > > We visualize ER instead of the Jacobian norm because ER is purely topological, while the Jacobian norm depends on model-specific factors such as parameter initialization and training, which are irrelevant to rewiring.
> > >
> > > For more details, see the new figures here: [Fig link](https://anonymous.4open.science/r/Submission3560_Rebuttal_for_Reviewer_geiv/Fig.pdf)
> > >
> > > ### Q5 & Q7.
> > >
> > > We make two key assumptions in our approximation:
> > > 1. **Assumption 1**: The latent graph has uniform edge weights.
> > > 2. **Assumption 2**: For an observed sparsifier $G$, the sampling frequency of an edge is proportional to its weight $w_e$.
> > >
> > > Since we begin with **densification**, the observed sparsified graph $G = (V, E)$ is unweighted, with $w_e = 1$; this allows us to simplify the expression by omitting the exponential term $w_e$ in Eq. (3). The resulting expression involves only the probabilities $1-(1-p_e)^q$ and $(1-p_e)^q$, which describe the presence or absence of edges in the sparsified graph after $q$ rounds of independent sampling.  Thus, we can ensure that our approximation $G_l=(V,E_l)$ and the optimal solution by the accurate MLE $G_*=(V,E_*)$ satisfy: $E_l=E_*$.
> > >
> > > **Summary**: $G_l$ is the optimal uniform-edge solution to MLE problem under assumption 2 if we first apply densification.
> > >
> > > ### Q6.
> > >
> > > Please refer to the link above.
> > >
> > > ### Q8.
> > >
> > > Yes, we use Lemma D.1 in Eq. (4) of the appendix:
> > >
> > > $$
> > > \Phi S \Phi = \frac{1}{q} \sum_{i=1}^q y_i y_i^T
> > > $$
> > >
> > > where $y_i = \frac{\Phi_{:,e}}{\sqrt{p_e}}$ represents the contribution of the $i$-th sampled edge to $\Phi S \Phi$. The sampling process is i.i.d, so the assumption that $y_1, ..., y_q$ are i.i.d. holds regardless of the probability definition.
> > >
> > > ### Q9.
> > >
> > > Yes, even when feature similarity is included, the definition of $\Phi$ remains unchanged, i.e., $\Phi_{e,e} = R_e$.
> > >
> > > Thus, we still have:
> > >
> > > $$
> > > \mathbb{E} [yy^T] = \sum_e p_e \frac{1}{p_e} \Phi_{:,e} \Phi_{:,e}^T = \Phi^2 = \Phi
> > > $$
> > >
> > > The concrete definition of $p_e$ is first used in Eq. (9) when deriving the bound on the norm of $y$, and we can define $p_e$ as $(1 + \gamma)R_e$ with any $\gamma$, once $(1 + \gamma) > 0$ and is upper-bounded. The proof sketch holds without modification except for Eq. (9) .
> > >
> > > For Q8 & Q9, you can also refer to Corollary 6 from Spielman & Srivastava (2008).
> > >
> > > Once again, we would like to thank the reviewer for their time and effort. We wish you the best of luck with your own submissions!
> > >
> > > ---
> > > [1] Di Giovanni, Francesco, et al. "On over-squashing in message passing neural networks: The impact of width, depth, and topology." ICML 2023.
> > >
> > > [2] Choi, Jeongwhan, et al. "PANDA: Expanded Width-Aware Message Passing Beyond Rewiring." ICML 2024
> > >
> > > [3] Black, Mitchell, et al. "Understanding oversquashing in gnns through the lens of effective resistance." ICML 2023

---

### Official Review · Reviewer_1faH · 2025-03-15

**Overall Recommendation:** 3

**Summary:**

This paper proposes a method to mitigate the over-squashing issue of GNNs while preserving the spectrum of the original graph. To this end, the authors propose a two-stage process of densification followed by sparsification, where both steps are discussed in detail. Experimental results demonstrate the competitiveness of the proposed method across a wide range of datasets against several baselines.

**==Post-rebuttal update==**

I would like to thank the authors for their final response which partially addressed my concerns. On acknowledging this and as an encouragement of the good work from them, I am raising my score to 3. However, I would strongly recommend the authors to further strengthen the motivation of spectrum preservation for future versions of the paper.

**Claims And Evidence:**

The authors claimed “..it is important to preserve graph spectrum during rewiring, since many learning tasks, such as clustering and semi-supervised learning on graphs, rely on spectral properties to ensure accurate results..” - I wonder if the authors can elaborate on why learning tasks may benefit from preserving spectral properties, given this is a main motivation?

The authors claimed “..these methods often introduce many additional edges to enhance connectivity, which increases the computational cost of learning on the rewired output graph. Moreover, a denser output graph increases the risk of oversmoothing..” - first I think methods such as SDRF can actually maintain the edge density via the addition/deletion mechanism; second I also don’t think the single factor of increased edge density would necessarily lead to over-smoothing being more likely. Both should be carefully rephrased.

The authors claimed “..while these approaches improve connectivity, they often substantially change the topology, overlooking the significance of retaining the graph structure..” - again I don’t think SDRF change the graph topology substantially, actually it preserves the degree distribution pretty well (according to Fig.4 of that paper).

The authors claimed “..rather than deterministically selecting edges based on predefined connectivity metrics, our method employs a probabilistic edge modification strategy” - to my knowledge SDRF also involves a stochastic mechanism to select which edge to add.

**Essential References Not Discussed:**

The references seem adequate to me.

**Experimental Designs Or Analyses:**

Can the authors explain why the performance of GOKU in Table 1 and 2 can be attributed to preserving spectral properties?

Actually, methods such as LASER can still perform quite well on some datasets (in Table 2) while distorting the spectrum more so than GOKU. Can the authors explain this more?

Why is SDRF not shown in Fig.2? As mentioned above, it can actually preserve the degree distribution (hence probably the spectrum) pretty well. If this is true, its average performance in Table 1 and 2 actually indicates preserving spectrum isn’t always contributing to better performance. Can the authors comment on this?

**Methods And Evaluation Criteria:**

In 3.2 the authors mentioned several advantages of the proposed framework over existing methods, which are however not very clear to me:
- why do we necessarily want to preserve spectral properties? This needs to be discussed in more detail with respect to specific learning tasks.
- why do we need to follow a two-stage approach of densification and sparsification? Can we not consider an optimisation problem that solves for an optimal graph topology at once? This needs to be better justified conceptually.
- Why is it desirable to choose edges to modify in a stochastic manner? What’s the benefits of doing this over a deterministic method?

The authors defined being “spectrally similar” in Definition 2.1 as having similar Laplacian quadratic form, but I am not sure if this actually preserves the Laplacian eigenvalues? I would imagine something like the spectral distance (norm of difference of the two vectors corresponding to the two sets of Laplacian eigenvalues) would be more appropriate as a measure of similarity.

**Other Comments Or Suggestions:**

The notations are confusing at times. For example, in some part of the paper the input graph is denoted as G while in other part denoted as G_d. Then expression such as \phi(G_d)=G makes it even more confusing.

**Other Strengths And Weaknesses:**

Strengths:
- the paper tackles a timely problem
- the proposed framework is conceptually interesting
- the paper is relatively well written and clear to follow

Weaknesses:
- the proposed framework can be better motivated and justified
- certain claims are not accurate and should be rephrased
- experimental results need more discussion in relation to core motivation

Please see detailed comments above.

**Questions For Authors:**

See above.

**Relation To Broader Scientific Literature:**

The paper contributes to a growing literature of GNN methods developed to address the over-squashing issue. The main differences from and advantages over the existing methods are, however, not very clear to me. See [Methods And Evaluation Criteria] for more details.

**Theoretical Claims:**

Theoretical claims seem correct to me.

---

> ### Author Rebuttal · Authors · 2025-03-31
>
> We sincerely thank the reviewer for their time, effort, and constructive comments. Below are our responses.
>
> > ### **Comment 1**: Why $x^TLx$ as spectral similarity & advantages over degree distribution in SDRF
>
> **Summary**: Our approach defines similarity via the Laplacian quadratic form, aligning with the graph spectral sparsification literature and covering **broader structural properties** beyond degree distribution.
>
> Consider two graph $G$ and $\\tilde{G}$ with $(1-\\epsilon)x^TLx \\leq x^T\\tilde{L}x \\leq (1+\\epsilon)x^TLx$, we have:
> 1. **Eigenvalue Similarity**: The relationship $(1 - \\epsilon) \\lambda _i \\leq \\tilde{\\lambda} _i \\leq (1 + \\epsilon) \\lambda _i$ holds[1].
> 2. **Graph Cut Similarity**: The graph cut, $\text{cut}(G, S) = \\sum W(u, v)$, where $u \\in S$, $v \\in V - S$, and $W(u, v)$ is edge weight. Since $x^T L x = \\sum W(u, v) (x(u) - x(v))$, setting $x$ as indicator (1 inside $S$, 0 outside) gives $(1 - \\epsilon)\\text{cut}(G, S) \\leq \\text{cut}(\\tilde{G}, S) \\leq (1 + \\epsilon)\\text{cut}(G, S)$.
> 3. **Degree Distribution Similarity**: For node $u$, setting $x_u = 1$ and $x_v = 0$ for $v \\neq u$, we get $x^T L x = \\text{deg}(u)$. In other words, $(1-\\epsilon)\\text{deg}(G)(u) \\leq \\text{deg}(\\tilde{G})(u) \\leq (1+\\epsilon)\\text{deg}_G(u)$ for all $u \\in V$.
> 4. **Eigenvector Similarity**: By matrix perturbation theory, the eigenvectors $v _i$ and $\\tilde{v} _i$ of $L$ and $\\tilde{L}$ are close, with $\\| v _i - \\tilde{v} _i \\|_2 \\leq \\frac{\\| L - \\tilde{L} \\|_2}{|\\lambda _i - \\tilde{\\lambda} _i|}$.
>
> See SDRF visualization: https://anonymous.4open.science/r/Submission3560_Rebuttal_for_Reviewer_1faH/Fig.pdf
>
> > ### **Comment 2**: Literature review regarding SDRF.
>
> We appreciate your valuable suggestions and will revise the paper to rephrase them more accurately. Thank you!
> 1. **SDRF also considers edge deletion**: We will clarify it to avoid confusion. We acknowledge SDRF does involve both edge addition and deletion, as noted in our paper: "Curvature-based methods (citing SDRF) optimize connectivity by adding and removing edges." A key difference is that **SDRF requires adding and removing nearly the same number of edges to preserve degree distribution**. In contrast, our method allows a large edge reduction while keeping spectral similarity using weighted edges.
> 2. **Edge density and over-smoothing.** We agree that increased edge density does not necessarily lead to over-smoothing, as connection patterns also play a role.
> 3. **Why stochastic manner to edit edges.** SDRF uses a stochastic mechanism to select edges for addition, and our method extends this by also randomly selecting edges for deletion. **We consider stochastic mechanism a means (not a goal)**. GOKU maintains spectral similarity even with a significant reduction in edge count ($0.5 \\leq \\beta \\leq 1$). This is made possible by spectral sparsification, which relies on stochastic sampling.
>
> > ### **Comment 3**: Spectrum and specific learning tasks & LASER performs well in some cases while not preserving spectrum well.
>
> 1. We summarize the **average similarity of eigenvalue distributions** between graphs from the same and different classes below. We use the fastdtw algorithm to compute distance between vectors of different sizes, and sim = $1/(0.01 + dis)$. It shows that **graphs from the same class exhibit more similar spectra**. Preserving spectrum could help classification tasks. Additionally, $x^TLx$ captures important graph anomaly signal, which **helps anomaly detection** [2].
> 2. Apart from spectrum, **others such as connectivity also matter**. While LASER distorts the spectrum more than GOKU, it also improves the spectral gap more. GOKU seeks to balance connectivity improvements with spectral similarity, as evidenced by the best overall performance.
>
> |  | Same class| Diff |
> | -- | --| --|
> | IMDB  | 0.59 | 0.21 |
> | MUTAG | 0.58| 0.23 |
> | PROTEINS | 0.13  | 0.05 |
> | ENZYMES | 0.05 | 0.03 |
>
> > ### **Comment 4**: Notations, $G$ and $G_d$ are confusing at times.
>
> $G_d$ is not an input to either the densification or sparsification processes, but a **candidate latent graph**. $G$ represents the input graph fed into the rewiring process. In densification, given $G$, we aim to find a latent graph $G_l$ that maximizes the probability as $G_l = \\arg \\max P(\\phi(G_d) = G)$, where $G$ is the actual input.
>
> > ### **Comment 5**: Why two-stage approach.
>
> The two-stage approach is **simple and practical**: in densification, when adding edges (Eq. 3, $\prod_{e \\in E} (1 - (1 - p_e)^q)^{w_e} \\prod_{e' \\in E'} ((1 - p_{e'})^q)^{\\bar{w}}$), we assume original edges $E$ are preserved, so we only need to find the $E'$ that maximizes the objective from the complement graph. However, allowing edge removal adds complexity, as we must also find an optimal subset from $E$ to delete.
>
> [1] Spectral sparsification of graphs: theory and algorithms.
>
> [2] Rethinking GNNs for Anomaly Detection.

---

> > ### Comment · Reviewer_1faH · 2025-04-04
> >
> > I thank the authors for addressing my previous concerns. With the clarifications they provide most things become clearer, especially the use the Laplacian quadratic form in assessing spectral preservation. The comparison with SDRF is also very helpful.
> >
> > However, I still have a main concern which is key to the paper's contribution: why does one need to necessarily preserve the spectrum in the first place?
> > - If the goal was to mitigate over-squashing, then improving sensitivity between nodes of long distance should be a priority, but it is not clear to me why preserving the spectrum can help with this. If anything, this can even be detrimental as there might be cases where it is necessary to modify the spectrum to improve connectivity (it would be interesting to check if this is the case for datasets where LASER achieves good performance while not preserving the spectrum).
> > - If the goal was to preserve spectrum to achieve good graph classification, as the authors clarified that graphs in the same class share similar spectrum (this is very good evidence to provide and I thank the authors for that), then why should one bother with an elaborate process of rewiring and not simply use the spectrum for classification (as is done in this paper https://arxiv.org/abs/1912.00735)?
> > - I understand there is a trade-off between these two goals (as the authors pointed out), but it is precisely this trade-off that is not clear to me given specific learning tasks. It would be more convincing if the authors can design synthetic tasks where they clearly demonstrate this, and use that to justify that the proposed method is necessary.
> > - What about node classification tasks? In this case it is not clear to me why preserving the spectrum would help, as the more important thing is the task at hand (eg node label distribution). There might be a similar trade-off there as well, but again this is not clear to me.
> >
> > In summary, while I agree the paper presents some interesting ideas that have potential, the motivation of spectral preservation needs to be strengthen in the first place. Without a convincing argument on this, one might feel this is a piece of work that proposes an interesting technical approach which also achieves good empirical performance on benchmarks, but its motivation and starting point is not sufficiently justified. With this improvement, however, I believe it would be a very good work.

---

> > > ### Author Response · Authors · 2025-04-05
> > >
> > > We thank the reviewer for their thoughtful and constructive suggestions. We fully agree on the importance of making the trade-off between mitigating over-squashing and preserving spectrum more transparent, which was a point we did not think of before, and we now provide both visualization and quantitative analysis to illustrate this.
> > >
> > > > **Settings**
> > >
> > > - **We evaluate LASER on graph classification tasks.** Recall that LASER identifies new edges by selecting node pairs $(i,j)$ with the smallest $A^k[i,j]$ where $A^k[i,j] > 0$ but $A[i,j] = 0$. To balance connectivity with spectrum preservation, we introduce a scoring function for each candidate edge:
> > > $$
> > > s = \alpha s_{\text{con}} + (1 - \alpha) s_{\text{spect}},
> > > $$
> > > where $s_{\text{con}}$ favors weakly-connected multi-hop neighbors, and $s_{\text{spect}}$ promotes spectrum preservation. Both scores are normalized to $[0, 1]$.
> > > This formulation generalizes LASER: **when $\alpha = 1$, we recover LASER; when $\alpha = 0$, we only consider spectrum preservation.**
> > >
> > > - We experiment on four graphs from the MUTAG dataset—two from class 1 (graphs 1 and 2) and two from class 2 (graphs 3 and 4). To assess the quality of rewiring under different $\alpha$, we define a **graph classification score**. For example, for graph 1:
> > > $$
> > > s_{1,\text{cls}} = \frac{ \|g_1 - g_3\| + \|g_1 - g_4\| }{ 2 \|g_1 - g_2\| },
> > > $$
> > > where $g_i = \text{MeanPool}(H_i)$ and $H = \text{GCN}(G_i, X_i)$ without a learnable weight matrix. A higher score implies that $g_1$ is more distinct from graphs of the opposite class (3 and 4) and closer to its classmate (graph 2), indicating better class separation.
> > > The overall classification score $s_{\text{cls}}$ is the average of all four graphs’ scores. We report the **percentage improvement** in $s_{\text{cls}}$ after rewiring to quantify the effect of different $\alpha$ values.
> > >
> > > > **Results**
> > >
> > > 1. We visualize the scores $s_{\text{con}}$ and $s_{\text{spect}}$ for a center node $i$ and compute the overall classification score $s_{\text{cls}}$ with $\alpha = 0, 0.5, 1.0$. Our results show that when $\alpha = 0.5$, the  $s_{\text{cls}}$ is the highest (best).
> > > 2. To further validate this, we use synthetic graphs to demonstrate situations where $\alpha = 0$ (spectrum preservation only) or $\alpha = 1$ (original LASER, improving connectivity only) is preferred. However, $\alpha = 0.5$ achieves the best overall performance.
> > >
> > > See figure and table: https://anonymous.4open.science/r/Submission3560_Rebuttal_for_Reviewer_1faH/Fig_and_table.pdf
> > > > **Summary**
> > >
> > > Although there is a tendency for certain scenarios to prefer connectivity or spectrum preservation, in real-world applications, **$\alpha = 0$ and $\alpha = 1.0$ are not generally optimal** since real-world datasets often exhibit mixed patterns not fully captured by any specific random graphs. A flexible rewiring method that can balance these two goals is necessary.
> > >
> > > ---
> > >
> > > > **Node classification**
> > >
> > > **Settings:**
> > >
> > > Given that $\lambda = \sum_{uv} A_{uv} (x_u - x_v)^2$, where $ x$ is the eigenvector associated with $\lambda$, we consider $x_u$ as the feature of node $u$ by concatenating components from the first 128 eigenvectors, **resulting in a 128-dimensional feature vector per node**. We then compute the average cosine similarity between nodes from the same class and different classes, as shown below:
> > >
> > > **Result:**
> > > | Dataset   |Same Class Sim|  Different Class Sim |
> > > |------|---|---|
> > > | Cora      | 0.2817   | 0.0736   |
> > > | Citeseer  | 0.0013  | 0.0001   |
> > > | Pubmed    | 0.3383    |0.1121   |
> > >
> > > **These results suggest that maintaining eigenvectors aids node classification tasks**. Additionally, preserving $x^T L x$ for $x\in R^n$ captures information important to node classification such as **degree and homophily ratio**. For example, to get homophily ratio of $u$, we can set $x_u=1/\sqrt{deg(u)}$, $x_v=0$ if $v$ has the same label as $u$, and $x_u=1/\sqrt{deg(u)}$, $x_v=1/\sqrt{deg(u)}$ otherwise $\forall v \in V$. In this case, $x^TLx=1$ if $u$ has homophily ratio 1.
> > >
> > > ---
> > >
> > > We would like to express our sincere gratitude to the reviewer again, particularly in reviewing the additional figures, tables, and discussion, which we understand require a significant investment of time. We hope that our response help address your concerns. Thank you!

---

### Decision · Program_Chairs · 2025-05-01

**Decision:**

Accept (poster)

**Comment:**

The paper introduces a spectrum-preserving graph sparsification approach to address over-squashing in GNNs. The paper has merits that the reviewers have highlighted. These include the methodological formalism and the good performance of the proposed approach, particularly when compared to existing pre-processing techniques. Nevertheless, the reviewers have also identified several limitations mainly related to the following points: (i) Motivation of simultaneously mitigating over-squashing while preserving the spectrum for node classification. This introduces a trade-off that hasn't been adequately discussed in the paper (and rebuttal). (ii) Tailoring experimental results to the main question of the paper. (iii) Besides, considering that the model acts in a pre-processing step, it's not evident how it compares to models that alleviate over-squashing while training the GNN. Considering that some of these points have been discussed in the response period, I'm in favor of accepting the paper. Please ensure that these comments are thoroughly addressed and incorporated into the manuscript.